# Substrate-specific effects of natural genetic variation on proteasome activity

**Mahlon A. Collins**●*, **Randi Avery**, **Frank W. Albert**●*

Department of Genetics, Cell Biology, and Development, University of Minnesota, Minneapolis, Minnesota, United States of America

* mahlon@umn.edu (MAC); falbert@umn.edu (FWA)

## Abstract

Protein degradation is an essential biological process that regulates protein abundance and removes misfolded and damaged proteins from cells. In eukaryotes, most protein degradation occurs through the stepwise actions of two functionally distinct entities, the ubiquitin system and the proteasome. Ubiquitin system enzymes attach ubiquitin to cellular proteins, targeting them for degradation. The proteasome then selectively binds and degrades ubiquitinated substrate proteins. Genetic variation in ubiquitin system genes creates heritable differences in the degradation of their substrates. However, the challenges of measuring the degradative activity of the proteasome independently of the ubiquitin system in large samples have limited our understanding of genetic influences on the proteasome. Here, using the yeast *Saccharomyces cerevisiae*, we built and characterized reporters that provide high-throughput, ubiquitin system-independent measurements of proteasome activity. Using single-cell measurements of proteasome activity from millions of genetically diverse yeast cells, we mapped 15 loci across the genome that influence proteasomal protein degradation. Twelve of these 15 loci exerted specific effects on the degradation of two distinct proteasome substrates, revealing a high degree of substrate-specificity in the genetics of proteasome activity. Using CRISPR-Cas9-based allelic engineering, we resolved a locus to a causal variant in the promoter of *RPT6*, a gene that encodes a subunit of the proteasome's 19S regulatory particle. The variant increases *RPT6* expression, which we show results in increased proteasome activity. Our results reveal the complex genetic architecture of proteasome activity and suggest that genetic influences on the proteasome may be an important source of variation in the many cellular and organismal traits shaped by protein degradation.

## Author summary

Protein degradation controls the abundance of cellular proteins and serves an essential role in protein quality control by eliminating misfolded and damaged proteins. In eukaryotes, most protein degradation occurs in two steps. The ubiquitin system first attaches ubiquitin to proteins to target them for degradation. The proteasome then selectively binds and degrades ubiquitinated proteins. Understanding how individual genetic

**Data Availability Statement:** Whole-genome sequencing data is available through the NIH Sequence Read Archive under Bioproject accession PRJNA885116. Computational scripts and data used for statistical analysis and to generate plots

are available at: http://www.github.com/mac230/proteasome_QTL_paper All remaining data are within the manuscript and its Supporting information files.

**Funding:** This work was supported by NIH grants F32-GM128302 to MAC and R35-GM124676 to FWA from the National Institute of General Medical Sciences (https://www.nigms.nih.gov/). The funders had no role in study design, data collection and analysis, decision to publish, or preparation of the manuscript.

**Competing interests:** The authors have declared that no competing interests exist.

differences affect the activity of the proteasome could improve our understanding of the many traits influenced by protein degradation. However, most assays that measure proteasomal protein degradation are not suitable for use in large samples or are affected by changes in the activity of the ubiquitin system. Using yeast, we built reporters that provide high-throughput measurements of proteasome activity independently of the ubiquitin system. We used measurements of proteasome activity from millions of live, single cells to identify regions of the genome with DNA variants that affect proteasomal protein degradation. We identified 15 such regions, showing that proteasome activity is a genetically complex trait. Using genome engineering, we found that one locus contained a variant in the promoter of a proteasome subunit gene that affected the activity of the proteasome towards multiple substrates. Our results demonstrate that individual genetic differences shape proteasome activity and suggest that these differences may contribute to variation in the many traits regulated by protein degradation, including gene expression, growth, development, aging, and disease.

## Introduction

Protein degradation helps maintain protein homeostasis by regulating protein abundance and eliminating misfolded and damaged proteins from cells. The primary protein degradation pathway in eukaryotes is the ubiquitin-proteasome system (UPS). The UPS consists of two functionally distinct components, the ubiquitin system and the proteasome [1–4]. Ubiquitin system enzymes target proteins for degradation by binding degradation-promoting signal sequences (termed "degrons" [5]) and covalently attaching chains of the small protein ubiquitin (Fig 1A) [2, 3, 6, 7]. The proteasome then degrades polyubiquitinated proteins using two elements, the 19S regulatory particle and the 20S core particle [1, 8, 9]. The 19S regulatory particle selectively binds polyubiquitinated proteins [4, 10] then deubiquitinates, unfolds, and translocates them to the 20S core particle, which degrades proteins to short peptides [11](Fig 1A). The UPS is responsible for 70–80% of intracellular protein degradation [4, 12] and influences the abundance of much of the proteome [13–15]. Therefore, UPS activity must be precisely and dynamically regulated at the levels of (1) substrate targeting by the ubiquitin system [16–18] and (2) proteasomal protein degradation [19, 20]. Imbalances between UPS activity and the proteolytic needs of the cell adversely impact cellular viability and are associated with a diverse array of human diseases, including cancers, immune disorders, metabolic syndromes, and neurodegenerative diseases [3, 20–23]. Determining the factors that create variation in substrate targeting by the ubiquitin system and proteasomal protein degradation could thus improve our understanding of the many traits influenced by protein degradation.

Until recently, it was largely unknown how natural genetic variation affects UPS protein degradation. To begin to address this question, we mapped genetic influences on the N-end Rule, a UPS pathway that recognizes degrons in protein N-termini (termed "N-degrons" [5, 24]). Our results showed that UPS activity is a genetically complex trait, shaped by variation throughout the genome [25]. Some of the largest genetic effects on N-end rule substrates resulted from variation in ubiquitin system genes. In particular, genes whose products process (*NTA1*) and recognize N-degrons (*UBR1* and *DOA10*) and ubiquitinate substrates (*UBC6*) each contained multiple causal variants that altered UPS activity, often in an N-degron-specific manner [25]. Thus, individual genetic differences in the ubiquitin system are an important source of substrate-specific variation in UPS protein degradation.

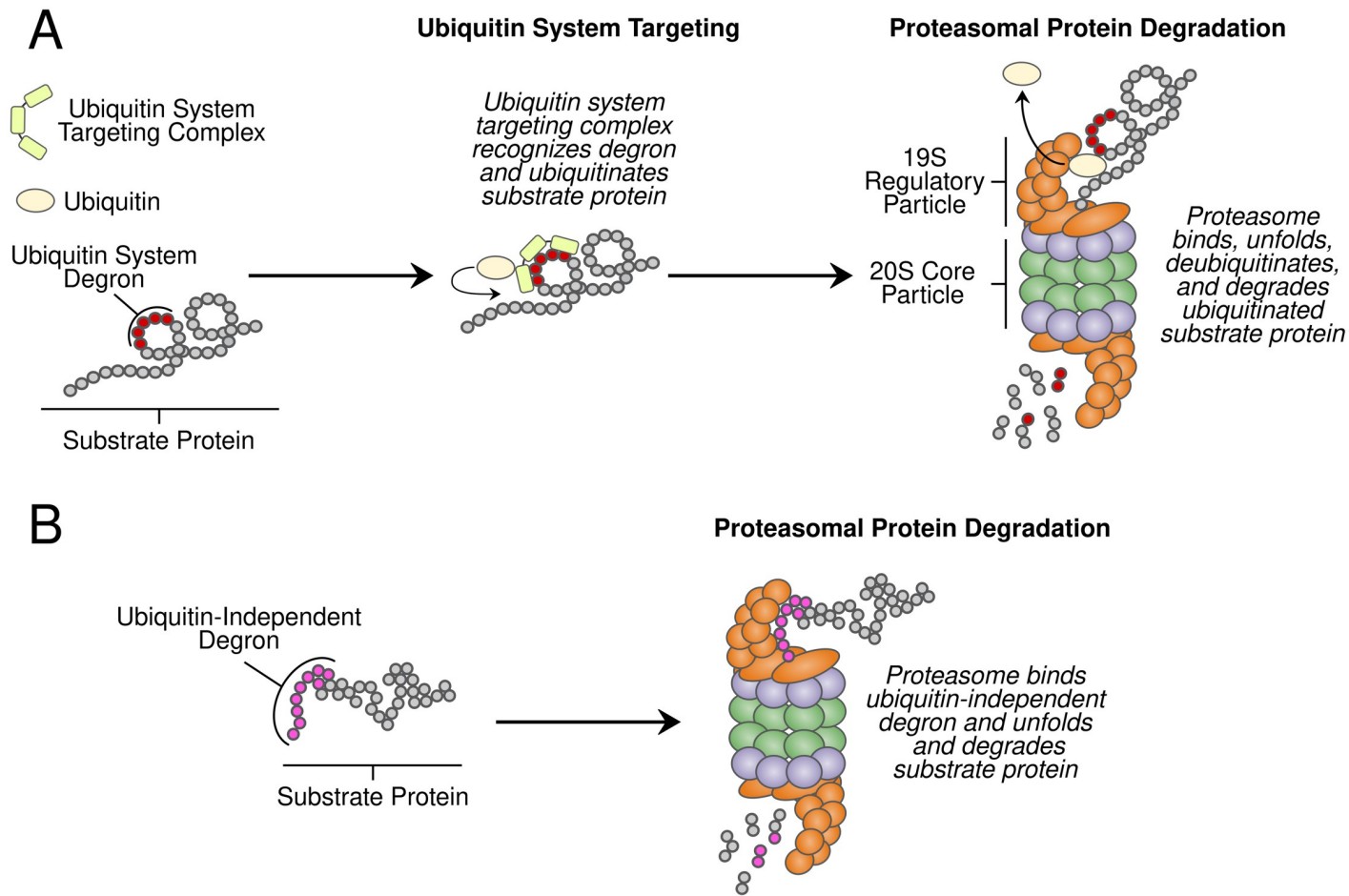

**Fig 1. Ubiquitin-dependent and -independent proteasomal protein degradation.** A. UPS protein degradation resulting from (1) ubiquitin system targeting followed by (2) proteasomal protein degradation. B. Proteins with ubiquitin-independent degrons are directly bound and degraded by the proteasome without ubiquitin system targeting.

We do not know whether genetic effects on the proteasome are as prominent as those on the ubiquitin system. Our understanding of how natural genetic variation influences proteasome activity is largely limited to the clinical consequences of variation in proteasome genes. Missense mutations in several proteasome genes that alter proteasome activity cause a spectrum of heritable disease phenotypes, including intellectual disability [26], lipodystrophy [27, 28], cataracts [29], recurrent fever [30], and morphological abnormalities [31]. Variation in proteasome genes has also been linked to multiple common diseases, including myocardial infarction [32], stroke [33], type 2 diabetes [34, 35], and cancer [36, 37]. However, these mutations and polymorphisms were identified through targeted sequencing of a subset of proteasome genes, leaving us with a biased, incomplete view of genetic influences on proteasome activity. Genome-wide association studies have linked variation in the vicinity of proteasome genes to a variety of organismal phenotypes [38–41]. However, these studies have neither fine-mapped these loci to their individual causal variants nor determined whether they alter proteasome activity.

A related question is whether variant effects on proteasome activity result in similar changes in the degradation of distinct proteasome substrates. Variation in protein half-lives spans

several orders of magnitude [42–44], in part as a result of proteasome-specific factors that are independent of the ubiquitin system, such as how readily proteins are bound, unfolded, and degraded by the proteasome. Substrate protein factors such as unstructured initiation region length [45–47], biases in amino acid composition [48–50], where in the protein degradation is initiated [45], and the stability of a protein's fold [48, 51] can all alter how readily a specific protein is degraded by the proteasome.

Proteasomes can also be assembled in multiple configurations that impart distinct affinities for different classes of substrates. In yeast, the catalytically active 20S core particle may be uncapped or singly or doubly capped with the 19S regulatory particle or the proteasome activator Blm10 [52]. 20S proteasomes capped with 19S regulatory particles have high affinity for polyubiquitinated proteins [53–55]. In contrast, uncapped 20S proteasomes preferentially degrade unfolded or intrinsically disordered substrates that have not been ubiquitinated [56–58]. Blm10-capped 20S proteasomes preferentially degrade short peptides, rather than proteins [59–61]. Genetic effects on the composition of the "proteasome pool" [62] could, therefore, create substrate-specific changes in protein degradation.

Technical challenges have precluded a more systematic understanding of the genetics of proteasomal protein degradation. The effects of natural DNA polymorphisms are often subtle, necessitating large sample sizes for detection. Statistically powerful genetic mapping of cellular traits such as proteasome activity requires assays that can provide quantitative measurements from thousands of individuals [63]. At this scale, *in vitro* biochemical assays of proteasome activity are impractical. Several synthetic reporter systems can measure UPS activity *in vivo* with high throughput [64–66]. However, the output of these reporters reflects the activities of both the ubiquitin system and the proteasome, potentially hindering detection of variants that specifically affect the proteasome. In particular, QTLs often span dozens of genes and report the composite signal of multiple linked variants. As a result, the effect of a causal variant may not be reflected in a QTL if its effect direction is opposite one or more additional causal variants with larger effects. Multiple lines of evidence suggest that QTL regions often contain multiple causal variants [7, 25, 67, 68]. In our previous work, four QTL regions we fine-mapped to causal ubiquitin system genes each contained multiple causal variants and were, along with QTLs containing *HAP1* and *MKT1*, the most frequently detected and largest effect size QTLs for the N-end Rule [25]. This suggests that variant effects on the ubiquitin system could mask or obscure specific effects on the proteasome when mapping with UPS activity reporters. Therefore, approaches that can directly and specifically measure proteasome activity are needed to understand the genetics of this trait.

The proteasome degrades a handful of endogenous cellular proteins without ubiquitination, providing a means of directly measuring proteasome activity independently of the ubiquitin system (Fig 1B). These proteins contain ubiquitin-independent degrons, short peptides that promote rapid proteasomal degradation without ubiquitination [69–73]. Ubiquitin-independent degrons simultaneously function as proteasome recognition elements that engage the 19S regulatory particle and unstructured initiation regions for 20S core particle degradation (Fig 1B) [70, 72–77]. The degradation-promoting effect of these peptides is transferable; conjugating a ubiquitin-independent degron to a heterologous protein converts it to a short-lived, ubiquitin-independent proteasome substrate [72, 73, 75, 77, 78]. This property has been leveraged to create genetically encoded, high-throughput reporters of proteasome activity whose readout is independent of ubiquitin system activity [70, 78, 79].

Here, we combined ubiquitin-independent degron-based proteasome activity reporters with our recently developed, statistically powerful mapping strategy to study the genetics of proteasome activity in the yeast *S. cerevisiae*. Our results reveal a polygenic genetic architecture of proteasome activity that is characterized by a high degree of substrate specificity. One locus

contained a causal variant that increased the expression of *RPT6*, a proteasome 19S subunit gene, while other regions contained candidate causal genes with no known links to UPS protein degradation. Our results show that individual genetic differences are an important source of variation in proteasome activity that may contribute to the complex genetic basis of the many cellular and organismal traits influenced by protein degradation.

## Results

### Single-cell measurements reveal heritable variation in proteasome activity

We sought to develop a reporter system capable of measuring proteasome activity independently of the ubiquitin system *in vivo* with high throughput and quantitative precision. To do so, we built a series of tandem fluorescent timers (TFTs), fusions of two fluorescent proteins with distinct spectral profiles and maturation kinetics [80, 81]. Our TFTs contained the faster-maturing green fluorescent protein (GFP) superfolder GFP [82] (sfGFP) and the slower-maturing red fluorescent protein (RFP) mCherry [83] (Fig 2A). The two fluorophores in the TFT mature at different rates and, as a result, the RFP / GFP ratio changes over time. If the TFT's degradation rate is faster than the RFP's maturation rate, the TFT's output, expressed as the $-\log_2$ RFP / GFP ratio, is directly proportional to its degradation rate (Fig 2B). The sfGFP / mCherry TFT can measure the degradation of substrates with half-lives ranging from several minutes to several hours [84], making it an ideal reporter system for studying short-lived proteasomal substrates. The TFT's output is also independent of the construct's expression level [84], making it possible to use TFTs in genetically diverse cell populations without confounding from genetic influences on reporter expression, which are expected in a genetically diverse cell population [14, 25, 84–87].

To relate the TFT's output to proteasome activity, we fused the ubiquitin-independent degrons from the mouse ornithine decarboxylase (ODC) and yeast Rpn4 proteins to our TFTs (Fig 2C). When expressed in yeast, the mouse ODC degron is recognized, bound, and degraded by the proteasome [69, 75, 78]. This property has previously been used to measure proteasome activity *in vivo* in yeast cells [88]. We fused amino acids 410 through 461 of mouse ODC to the TFT's C-terminus, consistent with the sequence requirements of the ODC degron [70], to create the ODC TFT (Fig 2C). The Rpn4 protein contains a ubiquitin-independent degron in amino acids 1 to 80 [72, 73]. We fused this sequence to the TFT's N-terminus to create the Rpn4 TFT (Fig 2C). We reasoned that the distinct degron positions (C- and N-terminal), sequences, recognition mechanisms, and inferred 19S regulatory particle receptors [70, 72, 89] would allow us to identify potential substrate-specific genetic effects on proteasome activity.

We characterized the ODC and Rpn4 TFTs in live, single cells by flow cytometry. We first evaluated the sensitivity of each TFT by comparing each TFT's output in the BY laboratory strain and a BY strain lacking the *RPN4* gene (hereafter "BY *rpn4Δ*"). *RPN4* encodes a transcription factor for proteasome genes and deleting *RPN4* reduces proteasome activity [71, 76, 90]. Deleting *RPN4* strongly reduced the output from the ODC and Rpn4 TFTs in BY *rpn4Δ* (mean difference versus BY = 1.71 and 0.53, respectively; t-test $p$ = 1.4e-6 and 1.6e-13, respectively; Fig 2D and 2E), showing that our TFTs provide sensitive *in vivo* measurements of proteasome activity. Consistent with previous reports [74, 77, 78], in the BY strain the ODC TFT was more rapidly degraded than the Rpn4 TFT (mean difference = 1.05; t-test $p$ = 6.9e-10; Fig 2D and 2E). Taken together, our results show that our TFTs provide quantitative, substrate-specific, *in vivo* readouts of proteasome activity.

To understand how natural genetic variation affects proteasome activity, we measured the output of the ODC and Rpn4 TFTs in two *Saccharomyces cerevisiae* strains. We compared BY,

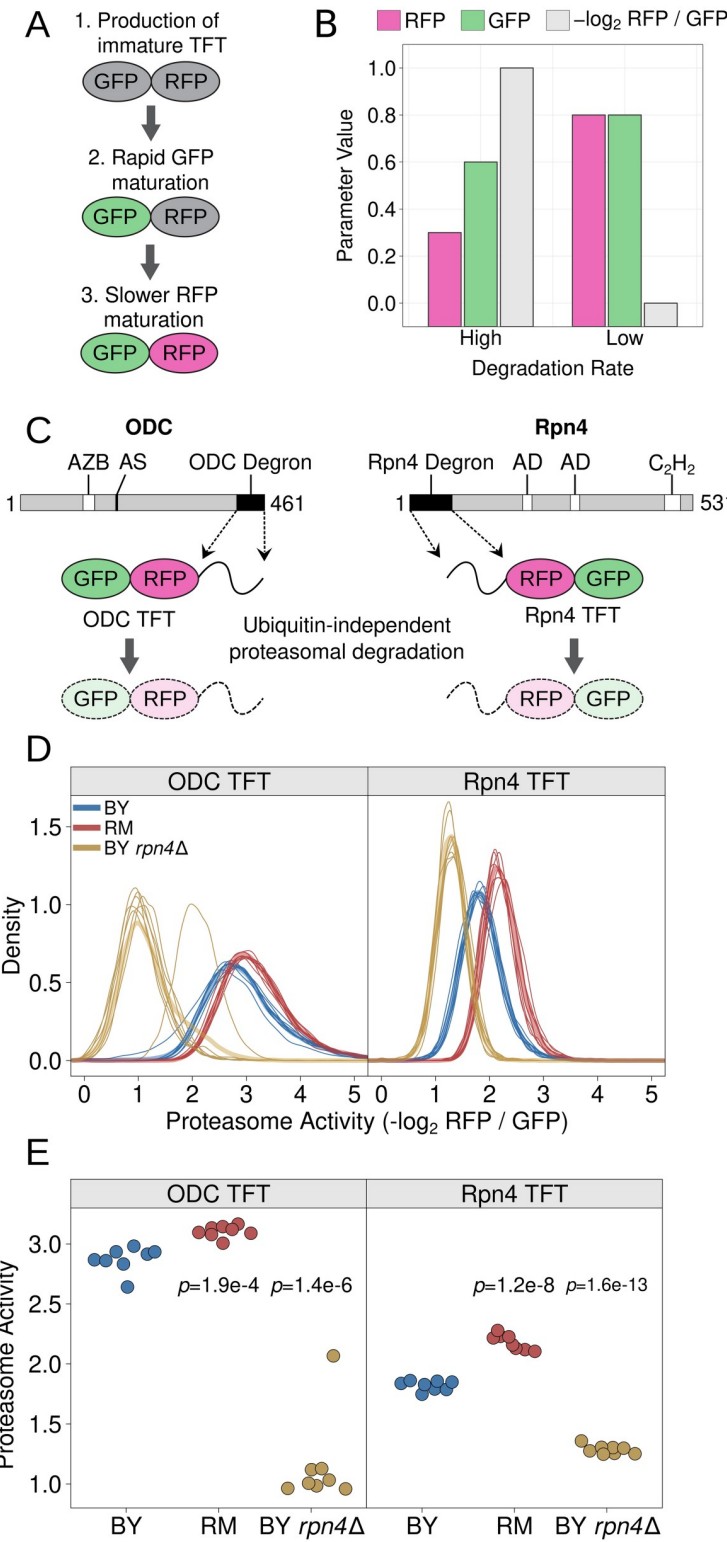

**Fig 2. Design and characterization of proteasome activity reporters.** A. Schematic of the production and maturation of a TFT. B. A bar plot created with simulated data showing how differences in a TFT's degradation rate influence the reporter's RFP and GFP levels, as well as the -log$_2$ RFP / GFP ratio. C. Diagram of mouse ODC and yeast Rpn4 showing the location of each protein's ubiquitin-independent degron. "AZB" = antizyme binding site, "AS" = active site, "AD" = transcriptional activation domain, "C$_2$H$_2$" = C$_2$H$_2$ zinc finger DNA binding domain. D. Density plots of

proteasome activity from 10,000 cells for each of 8 independent biological replicates per strain per reporter for the indicated strains and TFTs. Thin, opaque lines show individual biological replicates and thicker, transparent lines show the group average for the indicated strains. E. The median from each biological replicate in D. is plotted as a stripchart. t-test *p*-values are shown for the indicated strain versus BY.

which is closely related to the S288C reference strain, and the genetically divergent vineyard strain, RM, whose genome differs from BY at an average of 1 out of every 200 base pairs [91]. The RM strain showed higher proteasome activity towards the ODC and Rpn4 TFTs than BY (mean difference = 0.23 and 0.35, respectively; t-test *p* = 1.9e-4 and 1.2e-8, respectively; Fig 2D and 2E). We observed a significant interaction between strain background and proteasome substrate such that the magnitude of the BY / RM strain difference was greater for the Rpn4 TFT than the ODC TFT (two-way ANOVA interaction *p* = 0.013). Together, these results show that individual genetic differences create heritable, substrate-specific variation in proteasome activity.

## Bulk segregant analysis identifies complex, polygenic influences on proteasome activity

To map genetic influences on proteasome activity, we used our ODC and Rpn4 TFTs to perform bulk segregant analysis, a statistically powerful genetic mapping method that compares large numbers of individuals with extreme values for a trait of interest selected from a genetically diverse population [25, 86, 87, 92, 93]. In our implementation, the method identifies quantitative trait loci (QTLs), regions of the genome with one or more DNA variants that influence a trait, for proteasome activity. We created genetically diverse cell populations by mating BY strains harboring either the ODC or Rpn4 TFT with RM and sporulating the resulting diploids (Fig 3A). Using the resulting populations of haploid, genetically recombined progeny, we collected pools of 20,000 cells from the 2% tails of the proteasome activity distribution using fluorescence-activated cell sorting (FACS) (Fig 3B–3E). We then whole-genome sequenced each pool to determine the allele frequency difference between the high and low proteasome activity pools at each BY / RM DNA variant. At QTLs affecting proteasome activity, the allele frequencies will be significantly different between pools, while at unlinked loci the allele frequencies will be the same. We called significant QTLs using a logarithm of the odds (LOD) threshold previously determined to produce a 0.5% false discovery rate for TFT-based genetic mapping [25] (see "Materials and methods") and retained only QTLs detected at genome-wide significance in both of two independent biological replicates. We determined the direction of QTL effects by computing the difference in RM allele frequency between the high and low proteasome activity pools at each QTL peak position. When this value is positive, the RM allele of the QTL results in higher proteasome activity, while negative values indicate QTLs where the RM allele decreases proteasome activity. We identified 11 QTLs for the ODC TFT and 7 QTLs for the Rpn4 TFT (Fig 4, Tables 1 and S1). For the ODC TFT, the RM allele increased proteasome activity for 6 of 11 QTLs, while for the Rpn4 TFT, the RM allele increased proteasome activity for 2 of 7 QTLs. The distribution of proteasome activity QTL effect sizes, as reflected by the allele frequency difference between pools, was continuous and consisted predominantly of QTLs with small effects (Fig 4, Tables 1 and S1). Together, our mapping results demonstrate that proteasome activity is a polygenic trait, shaped by variation throughout the genome.

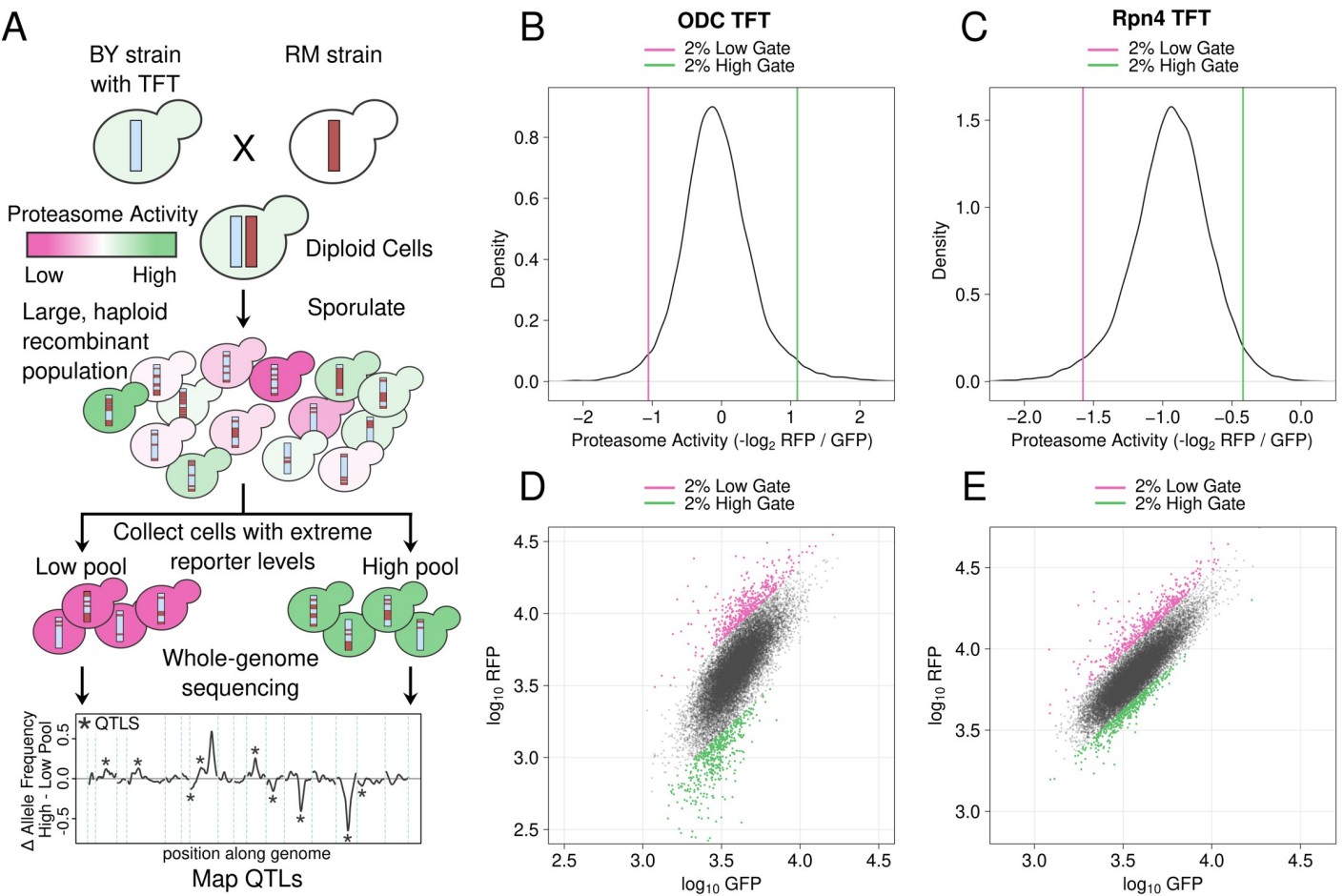

**Fig 3. Mapping genetic influences on proteasome activity using bulk segregant analysis.** A. Schematic of the experimental approach. B. / C. Proteasome activity distributions for the ODC TFT (B.) and Rpn4 TFT (C.). Vertical lines show the gates used to collect cells with extreme high or low proteasome activity. D. / E. Backplot of cells collected using the gates in B. / C. onto a scatter plot of GFP and RFP for the ODC (D.) and Rpn4 (E.) TFTs.

## Genetic influences on proteasome activity are predominantly substrate-specific

To study substrate specificity in the genetic architecture of proteasome activity, we evaluated the overlap in the sets of QTLs obtained with the ODC and Rpn4 TFTs. We defined overlapping QTLs as those whose peaks were within 100 kb of each other and that had the same direction of effect. We then calculated the overlap fraction for the two sets of QTLs by dividing the number of overlapping QTLs by the number of overlapping QTLs plus the non-overlapping QTLs for each reporter. Only three proteasome activity QTLs, V, VIIA, and XII, overlapped between the sets of QTLs detected with the ODC and Rpn4 TFTs (overlap fraction = 0.2, Fig 4, Tables 1 and S1), suggesting a high degree of substrate specificity.

To put this result in context, we examined overlap among our previously-described UPS N-end Rule activity QTLs [25]. The N-end Rule is divided into two primary branches based on how N-degrons are generated and recognized [94–97]. Based on the molecular mechanisms of Arg/N-degron processing and recognition, we hypothesized that QTLs affecting Arg/N-degrons would have predominantly substrate-specific effects. For example, 2 of 12 Arg/N-degrons require deamidation by Nta1 [97] and 4 of 12 Arg/N-degrons require arginylation by

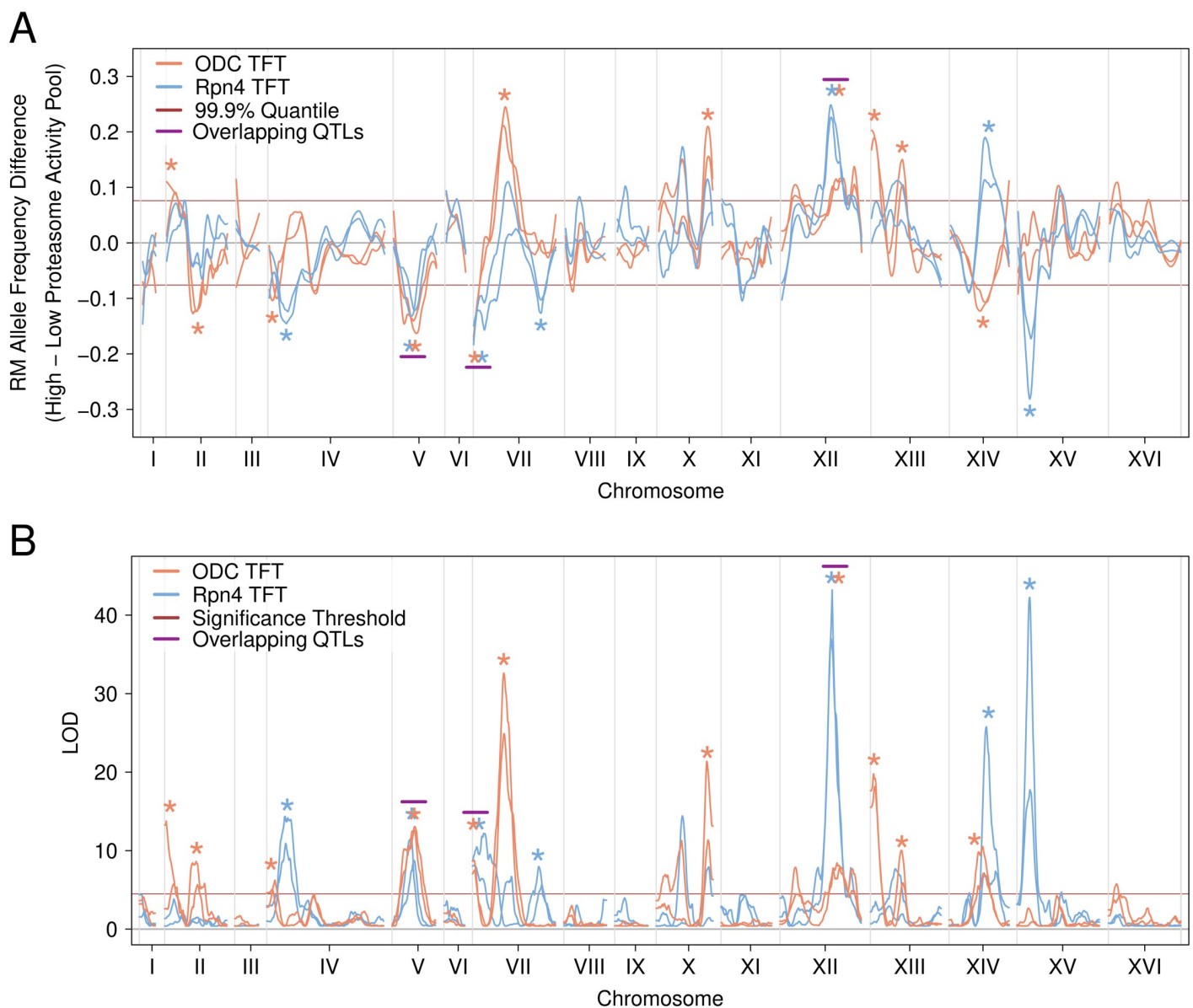

**Fig 4. Proteasome activity QTLs detected with the ODC and Rpn4 TFTs.** A. The line plot shows the loess-smoothed allele frequency difference between the high and low proteasome activity pools across the *S. cerevisiae* genome for each of two independent biological replicates per reporter. Asterisks denote QTLs, which are allele frequency differences exceeding an empirically-derived LOD score significance threshold (indicated in B.) in each of two independent biological replicates for a given reporter. The horizontal red lines denote an empirically-derived 99.9% quantile of the allele frequency difference. Magenta horizontal lines above pairs of asterisks denote QTLs detected with both TFTs with the same direction of effect, which are termed "overlapping QTLs". B. As in A., but for the LOD score for proteasome activity QTLs. The red horizontal line denotes the LOD score significance threshold used to call QTLs at a 0.5% FDR.

Ate1 [98]. QTLs affecting Nta1 or Ate1 would, therefore, be expected to influence, at most, 17% and 33% of Arg/N-degrons. Similarly, the E3 ligase Ubr1 has multiple binding sites and QTLs affecting these sites or their allosteric regulation [99] would also be expected to affect only a subset of Arg/N-degrons [25]. In contrast, Doa10, the E3 ligase for Ac/N-degrons, has a single RING-CH-type finger domain that has not been shown to differentially target distinct Ac/N-degrons. Similarly, the NatA complex acetylates 4 of 8 Ac/N-degrons [100], suggesting a

**Table 1. Proteasome activity QTLs detected with the ODC and Rpn4 TFTs.** The table lists all detected QTLs, sorted first by reporter, then by chromosome. Lowercase letters following chromosome numbers are used to distinguish QTLs on the same chromosome. "LOD", logarithm of the odds; "AFD", RM allele frequency difference (high proteasome activity pool minus low proteasome activity pool) at the QTL peak position. "Peak Position", "Left Index", and "Right Index" refer to base pair positions on the indicated chromosome. Each number is the average value calculated from two independent biological replicates for a given QTL.

| Reporter | Chromosome | LOD | AFD | Peak Position | Left Index | Right Index |
|---|---|---|---|---|---|---|
| ODC TFT | IIa | 9.76 | 0.10 | 69800 | 32850 | 107100 |
| ODC TFT | IIb | 7.13 | -0.12 | 418100 | 358850 | 462650 |
| ODC TFT | IVa | 5.64 | -0.10 | 85150 | 30400 | 127400 |
| ODC TFT | V | 12.83 | -0.15 | 291350 | 247700 | 325650 |
| ODC TFT | VIIa | 8.14 | -0.15 | 20000 | 0 | 52800 |
| ODC TFT | VIIb | 28.74 | 0.23 | 409000 | 390050 | 431700 |
| ODC TFT | X | 16.36 | 0.18 | 666850 | 649350 | 691550 |
| ODC TFT | XII | 8.13 | 0.11 | 768150 | 666200 | 846700 |
| ODC TFT | XIIIa | 18.96 | 0.19 | 47800 | 25200 | 75850 |
| ODC TFT | XIIIb | 7.96 | 0.13 | 410900 | 377350 | 450100 |
| ODC TFT | XIVa | 8.81 | -0.11 | 441750 | 381400 | 501600 |
| Rpn4 TFT | IVb | 12.64 | -0.13 | 240600 | 213200 | 309150 |
| Rpn4 TFT | V | 10.09 | -0.13 | 259650 | 218250 | 294900 |
| Rpn4 TFT | VIIa | 10.21 | -0.15 | 88550 | 53550 | 141350 |
| Rpn4 TFT | VIIc | 6.80 | -0.11 | 882500 | 840650 | 926150 |
| Rpn4 TFT | XII | 40.11 | 0.23 | 672850 | 661800 | 685750 |
| Rpn4 TFT | XIVb | 16.58 | 0.15 | 544150 | 497300 | 574600 |
| Rpn4 TFT | XV | 30.00 | -0.22 | 167400 | 142600 | 186200 |

potentially higher rate of QTL sharing for Ac/N-degrons compared to Arg/N-degrons. Consistent with these predictions, we previously showed that variation at *NTA1* and *UBR1* differentially affects Arg/N-degrons, while variation at *DOA10* and the Ac/N-degron E2 ubiquitin-conjugating enzyme *UBC6* affects Ac/N-degrons similarly [25]. To test if the distinct patterns of substrate specificity we observed in these specific examples extend to the full sets of QTLs detected for Arg/N-degrons and Ac/N-degrons, we computed the QTL overlap fraction among all pairs of Arg/N-degrons or Ac/N-degrons with at least 7 QTLs (to match the minimum number of TFT QTLs detected with an individual proteasome activity reporter) using the QTL overlap criteria above. QTLs for Ac/N-degrons overlapped across multiple reporters (median overlap fraction = 0.54; Fig 5A), while Arg/N-degron QTLs were more substrate specific (median overlap fraction = 0.21; Fig 5A). The distributions of overlap fractions for Arg/N-degrons and Ac/N-degrons were highly distinct (Fig 5A), making them an ideal reference to gauge the substrate-specificity of proteasome activity QTLs.

The overlap fraction for the two sets of proteasome activity QTLs (0.2) was close to the median overlap for Arg/N-degrons (0.21, Fig 5A). Thus, genetic influences on proteasome activity are as substrate-specific as those on N-degrons that are engaged by a broad variety of molecular mechanisms in the ubiquitin system [94]. Overlap among the two sets of proteasome activity QTLs was considerably lower than that for Ac/N-degrons (Fig 5A). Crucially, the current proteasome and previous N-end Rule QTLs were detected with a similar experimental design with similarly high statistical power. Therefore, these comparisons across datasets provide an estimate of substrate specificity that is immune to potential inflation from QTLs that truly affect multiple substrates but may appear to be substrate-specific because they happened to be detected with only one or a few reporters. The chromosome XIVa and XIVb QTLs, which occur at similar positions but have opposing effects on the degradation of the

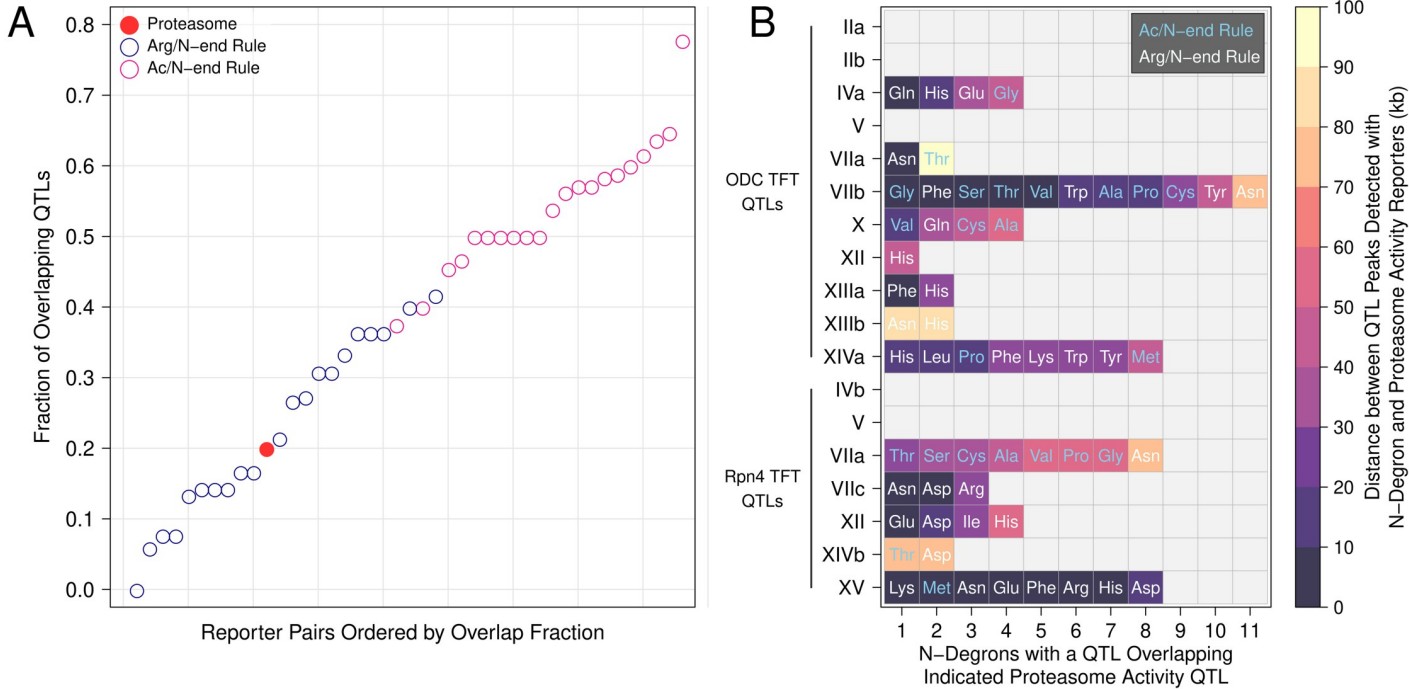

**Fig 5. Overlap of N-end Rule and proteasome activity QTLs.** A. Analysis of QTL overlap for proteasome activity, Arg/N-degron, and Ac/N-degron QTLs. For all pairs of reporters in the indicated reporter sets, we computed the overlap fraction as overlapping QTLs divided by total QTLs (overlapping QTLs plus reporter-specific QTLs). B. Overlap of proteasome activity and N-end Rule QTLs. The plot shows the number, identity, and N-end Rule branch of the N-degron QTLs that overlap proteasome activity QTLs. QTLs on the y axis are ordered first by reporter then by chromosomal position and labeled as in Table 1. N-degrons on the x axis are ordered by the distance of their QTL's peak position from the peak of the corresponding proteasome activity QTL detected with either the ODC or Rpn4 TFT.

Rpn4 and ODC TFTs (Fig 4A), provide further evidence that genetic effects on proteasome activity are highly substrate-specific.

## Effects of proteasome activity QTLs on the UPS N-end rule

We previously showed that four QTLs affecting the degradation of N-end Rule substrates contained causal variants in ubiquitin system genes [25]. As expected, these QTLs did not meet our criteria for overlap with any proteasome activity QTLs (S2 Table). However, many N-end Rule QTLs did not contain ubiquitin system genes, suggesting that they may result from genetic effects on processes unrelated to ubiquitin system targeting. To understand whether variation in N-end Rule activity could be explained by genetic effects on proteasome activity, we examined the overlap between the proteasome activity QTLs identified here and our previously-identified N-end Rule QTLs [25]. The set of N-end Rule QTLs comprises 149 QTLs detected with the 20 possible N-degron TFTs. However, many N-end Rule QTLs detected with distinct reporters overlap. To account for this, we applied our criteria for QTL overlap, which reduced the 149 N-end Rule QTLs detected with multiple reporters to 35 distinct, non-overlapping QTL regions. Eleven proteasome activity QTLs overlapped one of these 35 N-end Rule QTL regions (31%), suggesting that genetic effects on proteasome activity play a prominent role in shaping the activity of the UPS N-end Rule (Fig 5B).

Conversely, 4 of 15 proteasome activity QTLs did not overlap any N-end Rule QTLs, demonstrating that genetic variation can specifically alter the turnover of ubiquitin-independent proteasome substrates (Fig 5B). In particular, the chromosome V QTL altered the degradation

of both the ODC and Rpn4 TFTs, but no N-end Rule TFTs (Fig 5). These results suggest that multiple mechanisms can independently influence ubiquitin-independent versus ubiquitin-dependent proteasomal protein degradation.

## Overlapping proteasome activity and N-end rule QTLs identify candidate causal genes for proteasome activity

QTLs often span large intervals, complicating efforts to identify the underlying causal genes and variants. We reasoned that we could use overlapping proteasome activity and N-end Rule QTLs to more precisely estimate QTL peak positions and nominate candidate causal genes. To this end, we computed the overlaps between the sets of proteasome activity QTLs and N-end rule QTLs and used this information to identify candidate causal genes (Fig 5B). Two proteasome activity QTLs that were also detected with multiple N-degron TFTs occurred in genomic regions harboring variation that affects a multitude of traits in the BY / RM cross. The chromosome XIVa QTL was detected with the ODC TFT, 6 Arg/N-degron TFTs, and 2 Ac/N-degron TFTs (Fig 5B). The QTL's average peak position at base pair 462,767 was located approximately 4.5 kb from the *MKT1* gene. *MKT1* encodes a multifunctional RNA binding protein involved in 3' UTR-mediated RNA regulation [101, 102] and variation at *MKT1* affects multiple organismal traits in yeast, including sporulation efficiency and growth [103, 104]. The *MKT1* locus also occurs in a gene expression QTL "hotspot" that influences the expression of thousands of genes [85, 86] in the BY / RM cross. The chromosome XV QTL was detected with the Rpn4 TFT, 7 Arg/N-degron TFTs, and 1 Ac/N-degron TFT (Fig 5B). This set of QTL peaks clustered tightly at the average peak position of base pair 164,256. This position is approximately 7 kb away from *IRA2*, which encodes a negative regulator of RAS signaling [105]. Variation in *IRA2* affects the expression of thousands of genes in this cross of strains [106] via multiple causal variants that interact epistatically [67]. The QTL intervals for the chromosome XIVa and XV QTLs do not contain any genes encoding proteasome subunits or proteasome assembly factors. Therefore, the QTLs at *MKT1* and *IRA2* illustrate that some genetic effects on proteasome activity likely result from complex, indirect molecular mechanisms involving altered gene expression.

The chromosome VIIb QTL detected with the ODC TFT had the highest number of overlapping N-end rule QTLs, with QTLs detected in the same region with 4 Arg/N-degron and 7 Ac/N-degron TFTs (Fig 5B). The high number of overlapping N-end Rule QTLs for both Arg/N-degrons and Ac/N-degrons suggested that this QTL contained variation that broadly affects UPS protein degradation. The average chromosome VIIb QTL peak position at base pair 411,250 is within the *RPT6* open reading frame. *RPT6* encodes a subunit of the proteasome's 19S regulatory particle, suggesting that this QTL influences proteasome activity via direct effects on a proteasome subunit.

## Proteasome activity is shaped by a causal variant in the *RPT6* promoter

We selected the chromosome VIIb QTL for further experimental dissection. There are no missense *RPT6* variants between BY and RM. However, a single non-coding variant occurs at base pair 411,461 (Fig 6A) in an intergenic region between *RPT6* and the adjacent *ALG13*, which encodes an enzyme involved in oligosaccharide biosynthesis. We hypothesized that this intergenic variant (hereafter, "*RPT6* -175") was the causal nucleotide for the chromosome VIIb QTL.

To test the effect of *RPT6* -175 on proteasome activity, we used CRISPR-Cas9 to create BY strains with either the BY or RM alleles at *RPT6* -175. We tested the effect of the *RPT6* -175 RM allele on the ODC and Rpn4 TFTs, as well as a subset of Ac/N-degron and Arg/N-degron

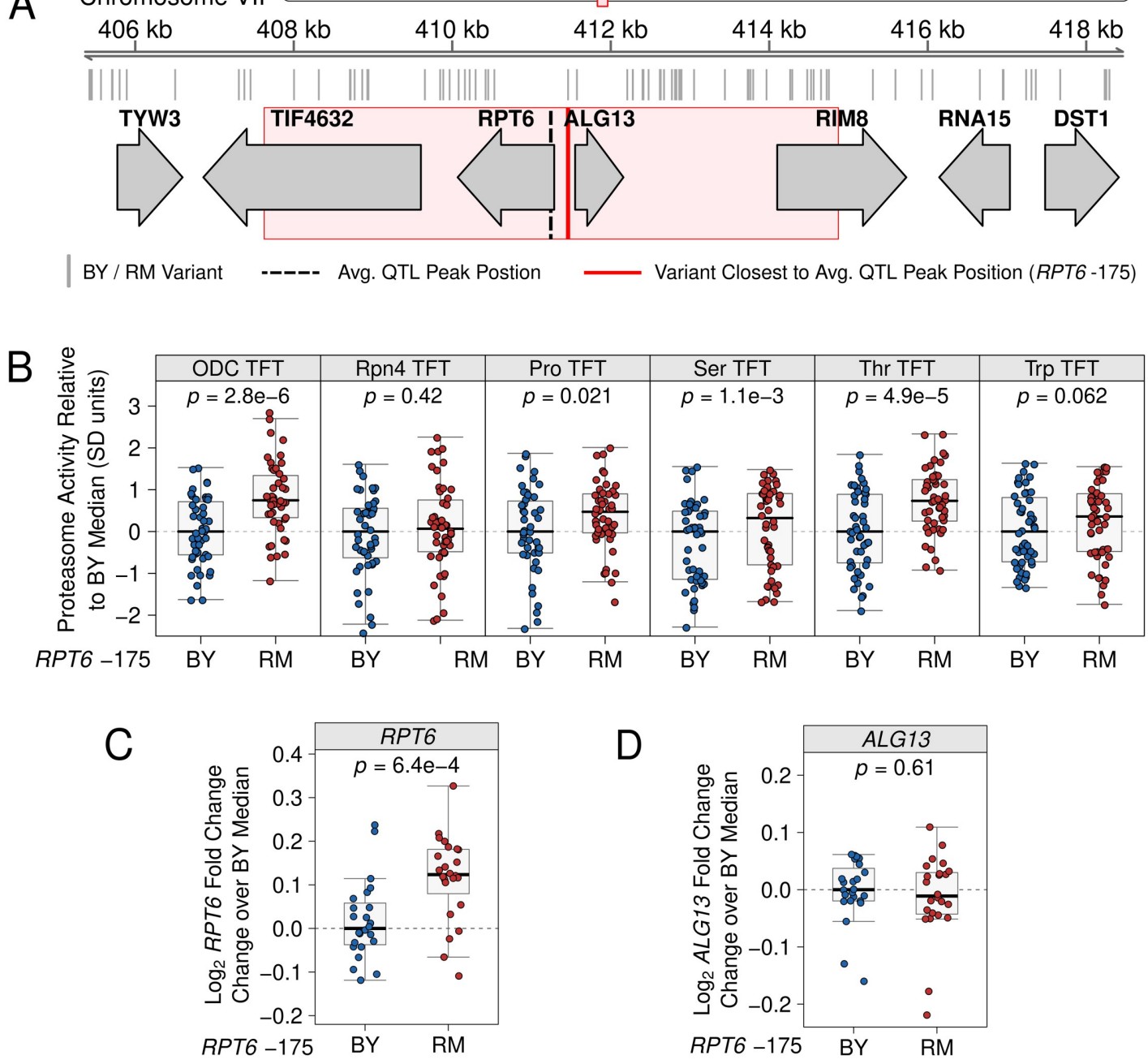

**Fig 6. Fine-mapping a causal variant for the chromosome VIIb QTL.** A. Genomic interval for the chromosome VIIb QTL. The red box depicts the 95% confidence interval of the chromosome VIIb QTL peak position, which was calculated using the chromosome VIIb QTL intervals from the ODC and N-end Rule TFTs with which the QTL was detected. B. CRISPR-Cas9 was used to engineer strains to contain either the BY or RM allele at *RPT6* -175 and the variant's effect on proteasome activity was measured by flow cytometry. The variant's effect was tested on strains harboring the ODC and Rpn4 ubiquitin-independent degron TFTs, as well as the proline (Pro), serine (Ser), and threonine (Thr) Ac/N-end TFTs, and the tryptophan (Trp) Arg/N-degron TFT. C. Rpt6 abundance was measured in strains harboring the BY or RM allele at *RPT6* -175. D. Alg13 abundance was measured in strains harboring the BY or RM allele at *RPT6* -175. In B-D, each point shows the median of 10,000 cells from independent biological replicates following Z-score normalization to the median of the control strain engineered to contain the BY allele at *RPT6* -175.

reporters with which the chromosome VIIb QTL was also detected. The *RPT6* -175 RM allele significantly increased proteasome activity towards the ODC TFT as compared to the BY *RPT6* -175 allele (mean difference = 0.79; $p$ = 2.8e-6; Fig 6B), but did not increase proteasome activity towards the Rpn4 TFT (mean difference = 0.17; $p$ = 0.42; Fig 6B). Among the tested N-degrons, the *RPT6* -175 RM allele significantly increased the degradation of the proline, serine, and threonine Ac/N-degron TFTs (mean difference = 0.44, 0.37, and 0.72, respectively; $p$ = 0.021, 1.1e-3, and 4.9e-5, respectively), while its effect on the degradation of the tryptophan Arg/N-degron was not statistically significant (mean difference = 0.26; $p$ = 0.062; Fig 6B).

We hypothesized that the RM allele of *RPT6* -175 increases proteasome activity by increasing *RPT6* expression. Increasing the expression of individual proteasome subunits is a well-established means of increasing proteasome activity [107–110]. In particular, increasing *RPT6* expression in human cells increases proteasome activity [110] and the turnover of proteasome substrates [111]. To understand the effect of the *RPT6* -175 RM allele on Rpt6 levels, we created yeast strains with the BY or RM allele at *RPT6* -175 and added an N-terminal tag encoding the green fluorescent protein mNeon to the chromosomal *RPT6* locus. Because *RPT6* -175 occurs in an intergenic region with putative promoters for *RPT6* and the divergently oriented *ALG13* (Fig 6A), we also created strains expressing an N-terminally mNeon-tagged Alg13 with the BY or RM allele at *RPT6* -175. We then used flow cytometry to measure Rpt6 and Alg13 expression. The *RPT6* -175 RM allele significantly increased Rpt6 abundance (median $\log_2$ fold change = 0.12; $p$ = 6.4e-4; Fig 6C), but did not affect Alg13 abundance (median $\log_2$ fold change = -0.01; $p$ = 0.61; Fig 6D). Therefore, *RPT6* -175 likely increases proteasome activity by increasing *RPT6* expression.

To better understand how *RPT6* expression levels influence proteasomal protein degradation, we measured how plasmid-based overexpression of *RPT6* affected proteasome activity. We created plasmids containing *RPT6* expressed from its native promoter or the strong, constitutively active *ACT1* promoter. To measure *RPT6*'s expression level, we created yeast strains in which both the plasmid and chromosomal copy of *RPT6* was tagged with mNeon and measured each gene's abundance using flow cytometry. As expected, plasmid-based overexpression *RPT6* led to a significant increase in *RPT6* expression compared to strains with an otherwise identical empty vector lacking the extra gene copy (fold change over empty vector = 1.55; $p$ = 1.7e-16; Fig 7A and 7B). Likewise, overexpressing *RPT6* from the *ACT1* promoter led to significant increases in the levels of Rpt6 above either strains with an empty vector or the native promoter overexpression plasmid (fold changes over empty vector and *RPT6* promoter = 3.47 and 2.09, respectively; $p$ = 1.6e-25 and 4.8e-19, respectively; Fig 7A and 7B). Together with our *RPT6* -175 edited strains, these plasmid overexpression strains thus allowed us to determine how Rpt6 abundance influences proteasome activity across a wide range of expression levels.

To measure the effect of *RPT6* overexpression on proteasome activity, we created plasmids to overexpress *RPT6* from its native promoter or the *ACT1* promoter, but without an mNeon tag. We built strains harboring one of these plasmids or an empty vector control plasmid as well as either the genomically-integrated ODC or Thr N-degron TFT and measured the degradation of each TFT by flow cytometry. Increasing *RPT6* expression either via the native *RPT6* or the *ACT1* promoter increased the degradation of both the ODC (mean difference = 1.14 and 0.83, respectively; $p$ = 3.9e-7 and 1.8e-4; Fig 7A) and Thr N-degron TFTs (mean difference = 0.94 and 1.11, respectively; $p$ = 7.7e-6 and 7.3e-7, respectively; Fig 7A), further suggesting that the effect of the causal *RPT6* -175 variant results from its effects on *RPT6* expression. The causal variant's effect on *RPT6* expression is small, even in the context of natural variants [68]. This raises the possibility that we may not have detected true *RPT6* -175 effects on the Rpn4 and Trp N-degron TFTs (Fig 6B) due to incomplete statistical power. Nevertheless, our

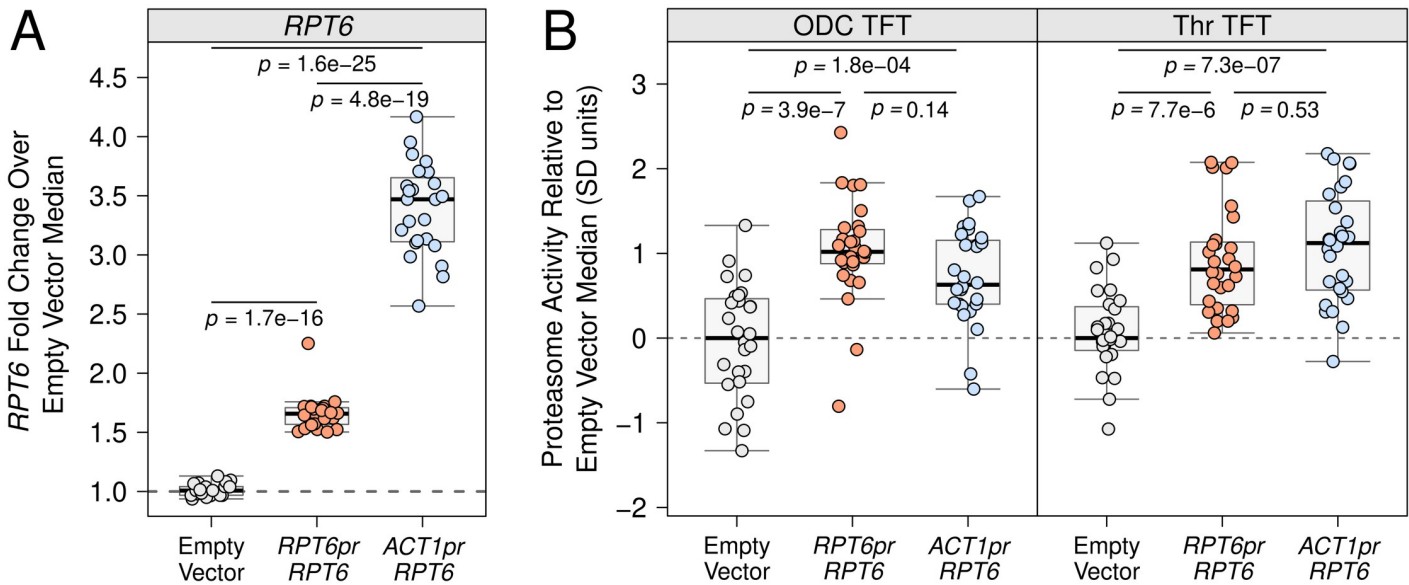

**Fig 7. Effect of *RPT6* overexpression on proteasome activity.** A. A BY strain with the chromosomal *RPT6* gene tagged with the GFP mNeon was transformed with plasmids containing the *RPT6* gene tagged with mNeon and expressed from the native *RPT6* promoter or the strong, constitutively active *ACT1* promoter. Rpt6 levels were measured by flow cytometry and compared to the same strain transformed with an otherwise identical empty vector control plasmid lacking the mNeon-tagged *RPT6* gene. B. BY strains harboring either the ODC or Thr N-degron TFTs were transformed with plasmids expressing *RPT6* without an mNeon tag from the native *RPT6* or *ACT1* promoters. Proteasome activity was measured by flow cytometry and compared to strains harboring an otherwise identical empty vector control plasmid lacking the *RPT6* gene. In A. and B. each point represents the median of 10,000 cells after Z-score normalization to the median of the corresponding empty vector control strain.

results further establish that increasing the expression of individual proteasome subunits generally and increasing Rpt6 specifically is sufficient to increase proteasome activity.

Interestingly, the significant increase in Rpt6 levels resulting from a second copy of *RPT6* driven by the *ACT1* versus *RPT6* promoter did not further enhance the degradation of either the ODC or Thr TFTs (mean difference = 0.31 and -0.17; *p* = 0.14 and 0.53, respectively; Fig 7B). Thus, by using multiple levels of *RPT6* overexpression, we have revealed limits to the extent to which increasing *RPT6* expression can drive increased proteasome activity.

To better understand the molecular properties, evolutionary history, and population characteristics of *RPT6* -175, we performed several additional analyses. To understand how *RPT6* -175 might increase *RPT6* expression, we scanned the *RPT6* promoter with either the BY or RM allele at *RPT6* -175 for putative transcription factor binding motifs. The *RPT6* promoter containing the RM, but not BY, allele at *RPT6* -175 contains a putative binding site for Yap1 (Fig 8A). Yap1 is a stress-associated transcription factor that indirectly increases proteasome activity during cellular stress, in part, by increasing expression of the proteasome gene transcription factor *RPN4* [112–114]. A multi-species alignment of the *RPT6* promoter, showed that the *RPT6* -175 BY allele is highly conserved among yeast species (Fig 8B). The BY allele is also present in the ancestral Taiwanese *S. cerevisiae* isolate, further indicating that the *RPT6* -175 RM allele is derived. We then examined *RPT6* -175 allelic status in a global panel of 1,011 *S. cerevisiae* isolates [115] to better understand its population characteristics and evolutionary origin. Across all strains, the *RPT6* -175 RM allele has a 33.7% population frequency (Fig 8C). However, among the "Wine / European" clade that contains RM, the *RPT6* -175 RM allele has a population frequency of 91.6% (Fig 8C). No other clades have a comparably high *RPT6* -175 RM allele frequency (Fig 8C). Thus, a derived variant whose ancestral allele is highly conserved

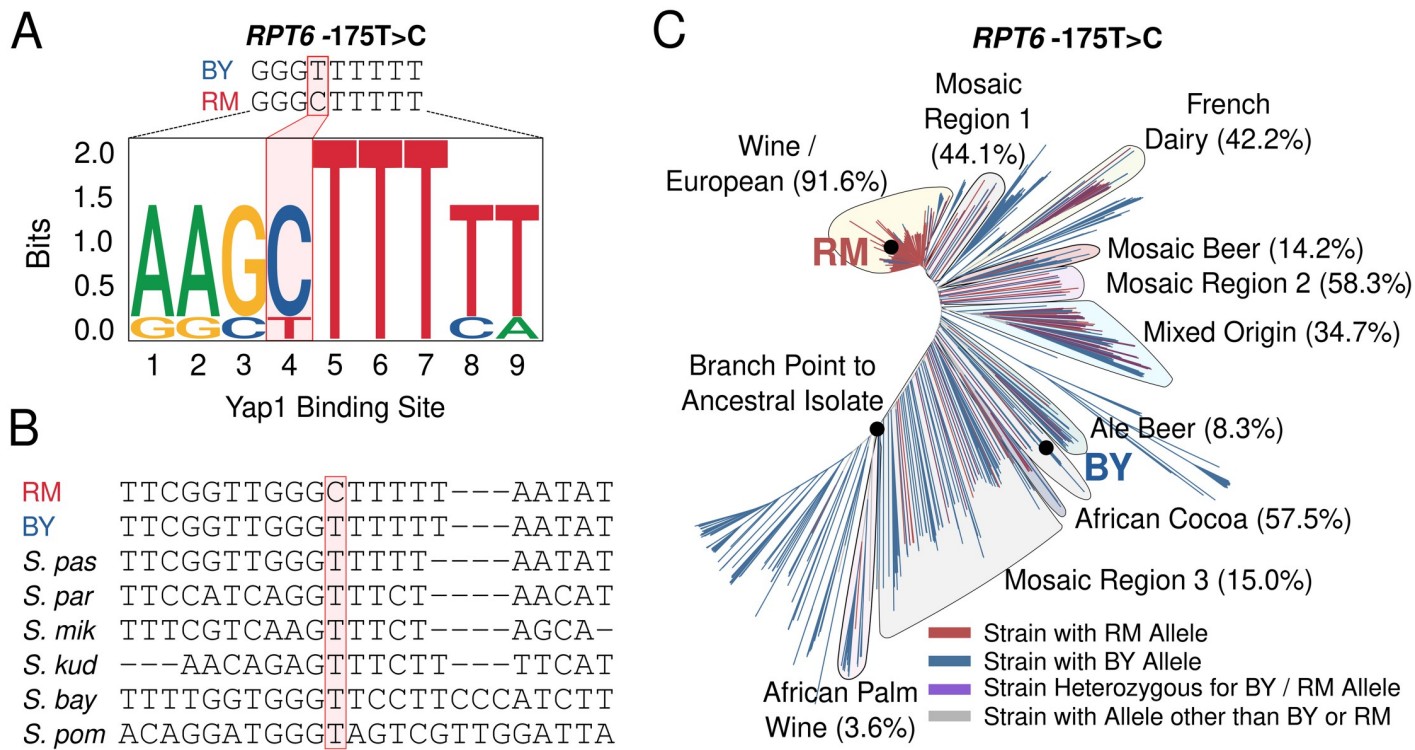

**Fig 8. Properties of the causal *RPT6* -175 causal variant.** A. The *RPT6* promoter with either the BY or RM allele at *RPT6* -175 was scanned for transcription factor binding motifs. The motif plot displays the sequence logo of a Yap1 binding motif created by the RM allele at *RPT6* -175. B. Multi-species alignment of the *RPT6* promoter with the causal -175 variant highlighted in red. *"S. pas"* = *Saccharomyces pastorianus*, *"S. par"* = *Saccharomyces paradoxus*, *"S. mik"* = *Saccharomyces mikatae*, *"S. kud"* = *Saccharomyces kudriavzevii*, *"S. bay"* = *Saccharomyces bayanus*, *"S. pombe"* = *Saccharomyces pombe*. C. Tree diagram showing the distribution of the *RPT6* -175 allele among a panel of 1,011 *S. cerevisiae* strains. Clades with the *RPT6* -175 RM allele are indicated along with its frequency in that clade in parentheses.

across yeast species can increase proteasome function, a result that stands in contrast to the often-deleterious consequences of new mutations.

## Discussion

Much of the proteome undergoes regulated turnover via proteasomal protein degradation [13–15]. Proteasome activity is thus a critical determinant of the abundance of individual proteins and, by extension, the functional state of the cell. Physiological variation in proteasome activity enables cells to adapt to changing internal and external environments, such as during cellular stress [107, 116, 117], while pathological variation in proteasome activity is linked to a diverse array of human diseases [3, 20, 23, 118]. However, a full understanding of the factors that determine proteasome activity has remained elusive. In particular, the challenges of measuring proteasomal protein degradation in large samples has limited our understanding of the genetics of proteasome activity. By combining high-throughput proteasome activity reporters with a statistically powerful genetic mapping method, we have established individual genetic differences as an important source of variation in proteasome activity. Our results add to the emerging picture of the complex effects of genetic variation on protein degradation, which include widespread effects on the activity of the ubiquitin system [25] and, as we show here, the proteasome.

This work provides several new insights into how individual genetic differences shape the activity of the proteasome. Previous studies identified rare mutations in proteasome genes as

the cause of a variety of monogenic disorders [27–29, 31, 118, 119]. However, it was unclear to what extent these mutations are representative of genetic effects on proteasome activity. Our results suggest that disease-causing mutations and disease-linked polymorphisms with large effects on proteasome activity represent one extreme of a continuous distribution of variant effects. Aberrant proteasome activity is a hallmark of numerous common human diseases [3, 20, 23]. Our results raise the possibility that the risk for these diseases may be subtly influenced by common variants that create heritable variation in proteasome activity. Our unbiased, genome-wide genetic mapping also identified QTLs containing no genes with previously-established connections to the regulation of proteasome activity. In particular, the chromosome XIVa and XV QTLs do not contain any genes encoding proteasome genes or proteasome assembly factors. Instead, the peaks of these QTLs occur near *MKT1* and *IRA2*, which encode an RNA-binding protein and a RAS signaling regulator respectively, further highlighting the complexity of genetic effects on proteasome activity and providing support for recent models of the genetics of complex traits, which emphasize the predominant role of weak, indirect *trans*-acting effects [120].

The proteasome activity QTLs we have identified add new insight into how genetic variation shapes the molecular effectors of cellular protein degradation. We recently mapped the genetics of the UPS N-end rule pathway and discovered multiple DNA variants that alter the activity of four functionally distinct components of the ubiquitin system [25]. Here, we extend this result by showing that genetic variation also shapes protein degradation through effects on the proteasome. Although many stimuli, such as protein misfolding or heat shock, cause coordinated changes in the activity of the ubiquitin system and the proteasome, recent work shows that these two systems can also be regulated independently and function autonomously of one another [19, 121]. For example, ubiquitination can initiate events besides proteasomal protein degradation, including lysosomal protein degradation, altered protein subcellular localization, and signaling cascade activation [121–123]. Likewise, a number of cellular proteins are bound and degraded by the proteasome without modification by the ubiquitin system [74]. Thus, predicting how genetic variation shapes the turnover of individual proteins will require consideration of genetic effects on both the ubiquitin system and the proteasome.

Our work further establishes the highly substrate-specific effects of natural genetic variation on protein degradation. This substrate specificity likely results from diverse mechanisms involving both direct and indirect effects on each step in the cascade of molecular events in UPS protein degradation, from substrate targeting by the ubiquitin system to proteasomal protein degradation. Our previous work [25] revealed that causal variants in ubiquitin system genes create direct, substrate-specific effects on UPS protein degradation by altering the sequence or expression of ubiquitin system genes whose products process, recognize, and ubiquitinate distinct sets of UPS substrates. We find a similarly high degree of substrate specificity among the set of proteasome activity QTLs (Figs 4 and 5), raising the question of how direct and indirect genetic mechanisms cause substrate-specific effects on the proteasome. Direct effects on the proteasome could arise through effects on substrate selection by the proteasome's 19S regulatory particle. Efficient degradation of the proteasome substrates tested here and in our previous study [25] require the proteasome's 19S regulatory particle [72, 124], which contains multiple substrate receptors. The Rpn4 degron is not bound by the canonical 19S receptors for polyubiquitin chains, Rpn10 [125] and Rpn13 [126], but instead is bound by Rpn2 and Rpn5 [72]. Genetic variation affecting 19S receptor abundance, affinity, or activity could therefore create direct, substrate-specific effects on the proteasome.

However, we think it is likely that many substrate-specific variant effects on protein degradation arise through indirect mechanisms. As an example, we mapped proteasome activity QTLs containing *MKT1* and *IRA2*, regions of the genome known to contain variation that

affects the expression of thousands of genes, including numerous UPS genes, in this cross of yeast strains [85, 87, 127–129]. Understanding how such highly pleiotropic QTL regions shape protein degradation is a difficult, but important challenge for future studies, particularly for understanding how genetic effects on protein degradation contribute to organismal traits, such as health, aging, and disease, traits that are likely shaped by the collective effects of many variants with small, indirect effects.

Based on the high degree of substrate specificity in the ubiquitin system and the proteasome [13, 14], we anticipate that the degradation of individual proteins will also be shaped by genetic effects that are highly substrate-specific. Understanding how natural genetic variation affects the proteome through effects on the degradation of individual proteins will thus require reporters that can sensitively measure the degradation of proteins with half-lives ranging from several minutes to several hours [42, 43]. The mCherry / sfGFP TFT is well-suited to this purpose. Previous studies have shown that this TFT should be capable of measuring the degradation of approximately 80% of yeast proteins based on their half-lives [84] and assuming the protein tolerates the TFT tag. Recently, a genome-wide TFT tagging approach successfully used the mCherry / sfGFP timer to measure the turnover of approximately 70% (around 4,000 proteins) of the yeast proteome [130], suggesting that degradation QTLs for most proteins could be mapped using this reporter. TFTs with red fluorescent proteins that mature over longer time scales, such as mRuby [131] or dsRed [132], could be used to measure the degradation of longer-lived proteins [84].

We identified QTLs for proteasome activity using bulk segregant analysis, an approach that has previously been used to characterize the genetic basis of variation in a host of molecular, cellular, and organismal traits [87, 91–93, 133]. By assaying large numbers of individuals, bulk segregant analysis provides high statistical power to detect variant effects on a trait [91, 92]. Here, we used high-throughput reporters to measure proteasome activity in millions of recombinant progeny from a cross of the BY and RM *S. cerevisiae* strains, which allowed us to reproducibly identify proteasome activity QTLs. Moreover, bulk segregant analysis is efficient in terms of time, labor, and resources as compared to linkage or association mapping. In particular, by generating two "bulks" with extreme phenotypes, we could detect proteasome activity QTLs through pooled whole-genome sequencing, rather than genotyping individual meiotic progeny. However, the choice of bulk segregant analysis also involves limitations that arise from using pooled whole-genome sequencing. Because we do not ascertain the genotypes of individual meiotic progeny, we cannot readily estimate the heritability of proteasome activity or the variance explained by the QTLs we detect. For the same reason, we are unable to detect genetic interactions between loci. Recent advances [134] could enable efficient, statistically powerful mapping of proteasome activity using individual meiotic progeny in future studies, which would address these limitations.

Using CRISPR-Cas9 based allelic engineering, we resolved a QTL on chromosome VII to a noncoding causal nucleotide that increases *RPT6* expression. Our results are consistent with previous studies in human cells, where overexpressing *RPT6* results in large increases in proteasome activity [110] and the turnover of proteasomal substrates [111]. Here, we observe similar effects by overexpressing *RPT6* in yeast, adding to a growing body of evidence that has established increasing proteasome subunit expression as a robust mechanism for increasing proteasome activity. In yeast, overexpression of the *SCL1* gene, which encodes the $\alpha$1 20S core particle, also increases proteasome activity and promotes resistance to cellular stress [107]. In *D. melanogaster*, overexpression of the *DME1* gene, which encodes the $\beta$5 subunit of the 20S core particle, increases proteasome activity and extends lifespan [135]. Similar effects occur in *C. elegans*, where overexpression of *PBS-5*, which encodes the $\beta$5 subunit [136], and the 19S subunit encoding *RPN-6* [137] each increase proteasome activity and promote resistance to

cellular stressors. In human cells, overexpression of 5 of 6 Rpt subunits (Rpt1-4 and Rpt6) of the 19S regulatory particle, *PSMA4*, which encodes the $\alpha$4 subunit of the 20S core particle, and *PSMD11*, which encodes the 19S subunit Rpn2, all increase proteasome activity [108–110].

While increasing proteasome subunit expression is thus an established means of increasing proteasome activity, the mechanism(s) of this effect are not well-understood. They may involve coordinated increases in the expression of additional proteasome genes or enhanced proteasome assembly. In the case of *RPT6*, one possibility is that increasing expression levels increases the number of 19S regulatory particles and, in turn, the fraction of 26S proteasomes. The proteasome pool comprises uncapped 20S core particle and 20S singly or doubly capped with 19S regulatory particles ("26S proteasomes") or other proteasome activators such as Blm10. Estimates of the fraction of uncapped 20S proteasomes in the proteasome pool vary across species and cell types, but are generally no less than 30% [58, 62, 138, 139], suggesting that a large fraction of the proteasome pool could be converted to 26S proteasomes. Moreover, the 26S fraction is dynamic and responsive to changes in 19S subunit expression. For example, in human cells, decreasing the expression of either of the 19S subunits Rpt6 and Rpn2 reduces the fraction of 26S proteasomes [140]. Current models of proteasome assembly posit that the 20S core particle can serve as a template for assembling the 19S regulatory particle [141–143]. Rpt6 plays a critical role in this process—insertion of its C-terminal tail into the $\alpha$2-$\alpha$3 pocket is the first step in assembling the 19S regulatory particle's base that sits atop the 20S core particle [142–144]. After insertion, Rpt6 functions as an anchor to which other RPT heterodimers are added [142–144]. These findings suggest that increasing *RPT6* expression could increase the 26S proteasome fraction by promoting the formation of an assembly intermediate that acts as a scaffold for further 19S assembly onto the 20S core.

Importantly, our measures of proteasome activity at multiple levels of *RPT6* overexpression suggest limits to the extent to which proteasome activity can be increased by overexpressing a single proteasome subunit (Fig 7). We observed no additional increase in proteasome activity when overexpressing *RPT6* from the *ACT1* versus *RPT6* promoter, despite the former producing a greater than 2-fold increase in Rpt6 abundance over the latter (Fig 7). Previous studies have reported increased expression of multiple proteasome genes [136] and more efficient proteasome assembly [137] in response to overexpression of single proteasome subunits. Potentially, these mechanisms become saturated at high levels of individual subunit overexpression, such that the overexpressed subunit is subject to degradation through quality control pathways that monitor subunit stoichiometry or other subunits become rate-limiting for proteasome or 19S assembly [145].

We have developed a generalizable strategy for mapping genetic effects on proteasomal protein degradation with high statistical power. The elements in our reporters function in many other eukaryotic organisms, including human cells [72, 75, 84]. Deploying the reporter systems developed here in genetically diverse cell populations may provide new insights into the genetic basis of a variety of cellular and organismal traits, including the many diseases marked by aberrant proteasome activity.

## Materials and methods

### Tandem Fluorescent Timer (TFT) proteasome activity reporters

We used TFTs, fusions of two fluorescent proteins with distinct spectral profiles and maturation kinetics, to measure proteasome activity. The most common TFT implementation consists of a faster-maturing green fluorescent protein (GFP) and a slower-maturing red fluorescent protein (RFP) [80, 81, 84, 130]. Because the two fluorescent proteins mature at different rates, the RFP / GFP ratio changes over time. If the TFT's degradation rate is faster than

the RFP's maturation rate, the negative $\log_2$ RFP / GFP ratio is directly proportional to the TFT's degradation rate [80, 84]. The RFP / GFP ratio is also independent of the TFT's expression level, [80, 84], enabling high-throughput, quantitative measurements of TFT turnover in genetically diverse cell populations [25, 84]. TFTs in the present study contained superfolder GFP (sfGFP) [82] and the RFP mCherry [83] separated by an unstructured 35 amino acid peptide sequence to minimize fluorescence resonance energy transfer [84].

To measure proteasome activity with our TFTs, we fused the ubiquitin-independent degrons from the mouse ornithine decarboxylase (ODC) and yeast Rpn4 proteins to our sfGFP-mCherry TFTs. ODC, an enzyme involved in polyamine biosynthesis, contains a ubiquitin-independent degron in its C-terminal 37 amino acids [69, 70, 78, 146]. Rpn4, a transcription factor for proteasome genes, contains a ubiquitin-independent degron in its N-terminal 80 amino acids [71, 72, 76]. Both degrons are recognized and bound by the 19S regulatory particle without ubiquitin conjugation and function as unstructured initiation regions [46] for 20S core particle degradation. Attaching either degron to a heterologous protein converts it into a short-lived proteasomal substrate. ODC degron-protein fusions have half-lives of approximately 6 minutes [147]. While the half-life of Rpn4 degron-protein fusions has not been precisely determined, previous results suggest it is between 10 and 20 minutes [72, 148].

We used a previously described approach [25] to construct TFT reporters and yeast strains harboring genomically integrated TFTs. Each TFT contained the constitutively active *TDH3* promoter, the *ADH1* terminator, codon-optimized sfGFP and mCherry, and the KanMX selection module that confers resistance to the antibiotic G418 [149]. TFTs were constructed so that the ubiquitin-independent degron was immediately adjacent to mCherry (Fig 2C), consistent with established guidelines for optimizing TFT function [81]. We used BFA0190 as the plasmid backbone for all TFT plasmids S3 Table. BFA0190 contains 734 bp of sequence upstream and 380 bp of sequence downstream of the *LYP1* ORF separated by a SwaI restriction site. We inserted TFT reporters into BFA0190 by digesting the plasmid with SwaI and inserting TFT components between the *LYP1* flanking sequences using isothermal assembly cloning (Hifi Assembly Cloning Kit; New England Biolabs [NEB], Ipswich, MA, USA). The 5' and 3' *LYP1* flanking sequences in each TFT plasmid contain natural SacI and BglII restriction sites, respectively. We produced linear DNA transformation fragments by digesting TFT-containing plasmids with SacI and BglII and gel purifying the fragments (Monarch Gel Extraction kit, NEB). Genomic integration of each linear transformation fragment results in deletion of the *LYP1* gene, allowing selection for TFT integration at the *LYP1* locus using the toxic amino acid analogue thialysine (S-(2-aminoethyl)-L-cysteine hydrochloride) [150–152] and G418 [149]. All plasmids used in this study are listed in S3 Table.

### Yeast strains and handling

**Yeast strains.**   We used two genetically divergent *Saccharomyces cerevisiae* yeast strains for characterizing our proteasome activity TFTs and mapping genetic influences on proteasome activity. The haploid BY strain used here (genotype: *MATa his3Δ hoΔ*) is a laboratory strain that is closely related to the *S. cerevisiae* S288C reference strain. The haploid RM strain used here is a vineyard isolate with genotype *MATα can1Δ::STE2pr-SpHIS5 his3Δ::NatMX AMN1-BY hoΔ::HphMX URA3-FY*. BY and RM differ, on average, at 1 nucleotide per 200 base pairs, such that approximately 45,000 single nucleotide variants (SNVs) between the strains can serve as markers in a genetic mapping experiment [86, 87, 91, 92]. We also engineered a BY strain lacking the *RPN4* gene (hereafter "BY *rpn4Δ*") to characterize the sensitivity and dynamic range of our TFT reporters. We replaced the *RPN4* gene with the NatMX cassette, which confers resistance to the antibiotic nourseothricin (clonNAT) [149]. To do so, we

**Table 2. Strain genotypes.**

| Short Name | Genotype | Antibiotic Resistance | Auxotrophies |
|---|---|---|---|
| BY | *MATa his3Δ hoΔ* | | histidine |
| RM | *MATα can1Δ::STE2pr-SpHIS5 his3Δ::NatMX hoΔ::HphMX* | clonNAT, hygromycin | histidine |
| BY *rpn4Δ* | *MATa his3Δ hoΔ rpn4Δ::NatMX* | clonNAT | histidine |

transformed BY with a DNA fragment created by PCR amplifying the NatMX cassette from plasmid from Addgene plasmid #35121 (a gift from John McCusker) using primers with 40 bp of homology to the 5' upstream and 3' downstream sequences of *RPN4* using the transformation procedure described below. To create strains in which the chromosomal copy of *RPT6* is N-terminally tagged with mNeon (Allele Biotechnology, San Diego, CA, USA), we PCR amplified the *RPT6* promoter from BY or RM genomic DNA, the mNeon open reading frame from plasmid BFA0129, the *RPT6* open reading frame from BY genomic DNA, and the KanMX resistance cassette from plasmid BFA0254. We then created transformation fragments containing these elements using splicing overlap extension PCR [153]. These DNA fragments were transformed into the BY strain using the procedure described below. Strain genotypes are presented in Table 2. S4 Table. lists the full set of strains used in this study.

The media formulations for all experiments are listed in Table 3. Synthetic complete media powders (SC -lys and SC -his -lys -ura) were obtained from Sunrise Science (Knoxville, TN, USA). We added the following reagents at the following concentrations to yeast media where indicated: G418, 200 mg / mL (Fisher Scientific, Pittsburgh, PA, USA); clonNAT (nourseothricin sulfate, Fisher Scientific), 50 mg / L; thialysine (S-(2-aminoethyl)-L-cysteine hydrochloride; MilliporeSigma, St. Louis, MO, USA), 50 mg / L; canavanine (L-canavanine sulfate, MilliporeSigma), 50 mg / L.

**Yeast transformations.** We used the lithium acetate / single-stranded carrier DNA / polyethylene glycol (PEG) method for all yeast transformations [154]. In brief, yeast strains were inoculated into 5 mL of YPD liquid medium for overnight growth at 30˚C. The next day, we diluted 1 mL of each saturated culture into 50 mL of fresh YPD and grew cells for 4 hours. Cells were washed in sterile ultrapure water and then in transformation solution 1 (10 mM Tris HCl [pH 8.0], 1 mM EDTA [pH 8.0], and 0.1 M lithium acetate). After each wash, we pelleted the cells by centrifugation at 3,000 rpm for 2 minutes in a benchtop centrifuge and

**Table 3. Media formulations.**

| Media Name | Abbreviation | Formulation |
|---|---|---|
| Yeast-Peptone-Dextrose | YPD | 10 g / L yeast extract |
| | | 20 g / L peptone |
| | | 20 g / L dextrose |
| Synthetic Complete | SC | 6.7 g / L yeast nitrogen base |
| | | 1.96 g / L amino acid mix -lys |
| | | 20 g / L dextrose |
| Haploid Selection | SGA | 6.7 g / L yeast nitrogen base |
| | | 1.74 g / L amino acid mix -his -lys -ura |
| | | 20 g / L dextrose |
| Sporulation | SPO | 1 g / L yeast extract |
| | | 10 g / L potassium acetate |
| | | 0.5 g / L dextrose |

discarded supernatants. After washing, cells were suspended in 100 $\mu$L of transformation solution 1 along with 50 $\mu$g of salmon sperm carrier DNA and approximately 1 $\mu$g of each linear transforming DNA fragment or approximately 300 ng of each transforming plasmid and incubated at 30˚C for 30 minutes with rolling. Subsequently, 700 $\mu$L of transformation solution 2 (10 mM Tris HCl [pH 8.0], 1 mM EDTA [pH 8.0], and 0.1 M lithium acetate in 40% PEG) was added to each tube, followed by a 30 minute heat shock at 42˚C. Transformed cells were then washed in sterile, ultrapure water, followed by addition of 1 mL of liquid YPD medium to each tube. Cells were incubated in YPD for 90 minutes with rolling at 30˚C to allow for expression of antibiotic resistance cassettes. We then washed the cells with sterile, ultrapure water and plated 200 $\mu$L of cells on solid SC -lys medium with G418 and thialysine for genomic integration of the TFTs, YPD plus G418 for chromosomal *RPT6* mNeon tagging, and YPD plus clonNAT for *RPT6* overexpression plasmids. We single-colony purified multiple independent colonies (biological replicates) from each transformation plate for further analysis as indicated in the text. Transformation at each targeted genomic locus was verified by colony PCR [155] using the primers listed in S5 Table.

**Yeast mating and segregant populations.** We used a modified synthetic genetic array (SGA) methodology [151, 152] to create populations of genetically variable, haploid recombinant cells ("segregants") for genetic mapping. BY strains with either the ODC or Rpn4 TFT were mixed with the RM strain on solid YPD medium and grown overnight at 30˚C. We selected for diploid cells (successful BY / RM matings) by streaking mixed BY / RM cells onto solid YPD medium containing G418, which selects for the KanMX cassette in the TFT in the BY strain, and clonNAT, which selects for the NatMX cassette in the RM strain. Diploid cells were inoculated into 5 ml of liquid YPD and grown overnight at 30˚C. The next day, cultures were washed with sterile, ultrapure water, and resuspended in 5 mL of SPO liquid medium (Table 3). We induced sporulation by incubating cells in SPO medium at room temperature with rolling for 9 days. After confirming sporulation by brightfield microscopy, we pelleted 2 mL of cells, which were then washed with 1 mL of sterile, ultrapure water, and resuspended in 300 $\mu$L of 1 M sorbitol containing 3 U of Zymolyase lytic enzyme (United States Biological, Salem, MA, USA) to degrade ascal walls. Asci were digested for 2 hours at 30˚C with rolling. Spores were then washed with 1 mL of 1 M sorbitol, vortexed for 1 minute at the highest intensity setting, and resuspended in sterile ultrapure water. We confirmed the release of cells from asci by brightfield microscopy and plated 300 $\mu$l of cells onto solid SGA medium containing G418 and canavanine. This media formulation selects for haploid cells with (1) a TFT via G418, (2) the *MATa* mating type via the *Schizosaccharomyces pombe HIS5* gene under the control of the *STE2* promoter (which is only active in *MATa* cells), and (3) replacement of the *CAN1* gene with *S. pombe HIS5* via the toxic arginine analog canavanine [151, 152]. Haploid segregants were grown for 2 days at 30˚C and harvested by adding 10 mL of sterile, ultrapure water and scraping the cells from each plate. Each segregant population cell suspension was centrifuged at 3000 rpm for 10 minutes and resuspended in 1 mL of SGA medium. We added 450 $\mu$L of 40% (v / v) sterile glycerol solution to 750 $\mu$L to each segregant culture and stored this mixture in screw cap cryovials at -80˚C. We stored 2 independent sporulations each of the ODC and Rpn4 degron TFT-containing segregants (derived from our initial matings) as independent biological replicates.

## Flow cytometry and fluorescence-activated cell sorting

**Flow cytometry.** We characterized our proteasome activity TFTs using flow cytometry. For all flow cytometry experiments, we inoculated yeast strains into 400 $\mu$L of liquid SC -lys medium with G418 for overnight growth in 2 mL 96 well plates at 30˚C with 1000 rpm mixing

**Table 4. Flow cytometry and FACS settings.**

| Parameter | Laser Line (nm) | Laser Setting (V) | Filter |
|---|---|---|---|
| forward scatter (FSC) | 488 | 500 | 488 / 10 |
| side scatter (SSC) | 488 | 275 | 488 / 10 |
| sfGFP / mNeon | 488 | 500 | 525 / 50 |
| mCherry | 561 | 615 | 610 / 20 |

on a MixMate (Eppendorf, Hamburg, Germany). The next day, 4 $\mu$L of each saturated culture was inoculated into a fresh 400 $\mu$L of G418-containing SC -lys media and cells were grown for an additional 3 hours prior to flow cytometry. We performed all flow cytometry experiments on an LSR II flow cytometer (BD, Franklin Lakes, NJ, USA) equipped with a 20 mW 488 nm laser with 488 / 10 and 525 / 50 filters for measuring forward and side scatter and sfGFP fluorescence, respectively, as well as a 40 mW 561 nm laser and a 610 / 20 filter for measuring mCherry fluorescence. Table 4 lists the parameters and settings for all flow cytometry and fluorescence-activated cell sorting (FACS) experiments.

All flow cytometry data was analyzed using R [156] and the flowCore R package [157]. We filtered each flow cytometry dataset to exclude all events outside of 10% ± the median forward scatter (a proxy for cell size). This gating approach captured the central peak of cells in the FSC histogram and removed cellular debris, aggregates of multiple cells, and restricted our analyses to cells of the same approximate size [25].

For flow cytometry experiments related to reporter characterization, we recorded 10,000 cells each from 8 independent biological replicates per strain for the ODC and Rpn4 degron TFTs. We extracted the median from each independent biological replicate and used these values for statistical analyses. The statistical significance of between strain differences for the ODC and Rpn4 degron TFTs was assessed using a two-tailed t-test without correction for multiple testing. We used an ANOVA with strain (BY or RM) and reporter (ODC or Rpn4 degron TFT) as fixed factors to assess the statistical significance of the interaction of genetic background with reporter.

For flow cytometry experiments related to fine-mapping the chromosome VIIb QTL, we used the following procedures. We recorded 10,000 cells each from 12 independent biological replicates per strain (BY *RPT6* -175 BY and BY *RPT6* -175 RM) per guide RNA per reporter (ODC and Rpn4 TFTs, as well as proline, serine, threonine, and tryptophan N-degron TFTs). We observed that, consistent with previous results [25], the output of the TFTs changed over the course of each flow cytometry experiment. We used a previously-described approach in which the residuals of a regression of the TFT's output on time were used to correct for this effect [25, 87]. We then Z-score normalized the sets of median values for each reporter, setting the mean equal to the median of a control BY strain engineered to contain the BY *RPT6* -175 allele. The effect of the *RPT6* -175 genotype was assessed using a linear mixed model implemented in the R packages 'lme4' [158] and 'lmertest' [159] using *RPT6* -175 genotype and guide RNA as fixed effects and plate as a random effect. We used a similar approach to measure mNeon tagged Rpt6 or Alg13 abundance in strains engineered to contain either the BY or RM allele at *RPT6* -175, except that the statistical significance of the difference between strains was assessed using a t-test uncorrected for multiple testing. For experiments measuring proteasome activity during *RPT6* overexpression, we evaluated statistical significance between strains using a linear mixed model with plasmid (empty vector, *RPT6* overexpression via the *RPT6* promoter, or *RPT6* overexpression via the *ACT1* promoter) as a fixed factor and plate as a random effect with Benjamini-Hochberg correction of *p* values [160].

**Fluorescence-Activated Cell Sorting (FACS).** We used FACS to collect pools of segregant cells for genetic mapping by bulk segregant analysis [86, 87]. We thawed and inoculated segregant populations into 5 mL of SGA medium containing G418 and canavanine for overnight growth at 30°C with rolling. The following morning, we diluted 1 mL of cells from each segregant population into a fresh 4 mL of SGA medium containing G418 and canavanine. Diluted segregant cultures were grown for 4 hours prior to sorting on a FACSAria II cell sorter (BD). Plots of side scatter (SSC) height by SSC width and forward scatter (FSC) height by FSC width were used to remove doublets from each sample and cells were further filtered to contain cells within ± 7.5% of the central FSC peak. We empirically determined that this filtering approach excluded cellular debris and aggregates while retaining the primary haploid cell population. We also defined a fluorescence-positive population by retaining only those TFT-containing cells with sfGFP fluorescence values higher than negative control BY and RM strains without TFTs. We collected pools of 20,000 cells each from the 2% high and low proteasome activity tails (Fig 3B–3E) from two independent biological replicates for each TFT. Pools of 20,000 cells were collected into sterile 1.5 mL polypropylene tubes containing 1 mL of SGA medium that were grown overnight at 30°C with rolling. After overnight growth, we mixed 750 $\mu$L of cells with 450 $\mu$L of 40% (v / v) glycerol and stored this mixture in 2 mL 96 well plates at −80°C.

## Genomic DNA isolation, library preparation, and whole-genome sequencing

To isolate genomic DNA from sorted segregant pools, we first pelleted 800 $\mu$L of each pool by centrifugation at 3,700 rpm for 10 minutes. Supernantants were discarded and cell pellets were resuspended in 800 $\mu$L of a 1 M sorbitol solution containing 0.1 M EDTA, 14.3 mM $\beta$-mercaptoethanol, and 500 U of Zymolyase lytic enzyme (United States Biological) to digest cell walls. Zymolyase digestions were carried out by resuspending cell pellets with mixing at 1000 rpm for 2 minutes followed by incubation for 2 hours at 37°C. After completing the digestion reaction, we pelleted and resuspended cells in 50 $\mu$L of phosphate-buffered saline. We then used the Quick-DNA 96 Plus kit (Zymo Research, Irvine, CA, USA) to extract genomic DNA according to the manufacturer's protocol, including an overnight protein digestion in a 20 mg / mL proteinase K solution at 55°C prior to loading samples onto columns. DNA was eluted from sample preparation columns using 40 $\mu$L of DNA elution buffer (10 mM Tris-HCl [pH 8.5], 0.1 mM EDTA). DNA concentrations for each sample were determined with the Qubit dsDNA BR assay kit (Thermo Fisher Scientific, Waltham, MA, USA) in a 96 well format using a Synergy H1 plate reader (BioTek Instruments, Winooski, VT, USA).

We used genomic DNA from our segregant pools to prepare a short-read library for whole-genome sequencing on the Illumina Next-Seq platform using a previously-described approach [25, 86, 87]. The Nextera DNA library kit (Illumina, San Diego, CA, USA) was used according to the manufacturer's instructions with the following modifications. We fragmented and added sequencing adapters to genomic DNA by adding 5 ng of DNA to a master mix containing 4 $\mu$L of Tagment DNA buffer, 1 $\mu$L of sterile molecular biology grade water, and 5 $\mu$L of Tagment DNA enzyme diluted 1:20 in Tagment DNA buffer and incubating this mixture on a SimpliAmp thermal cycler (Thermo Fisher Scientific) using the following parameters: 55°C temperature, 20 $\mu$L reaction volume, 10 minute incubation. We PCR amplified libraries prior to sequencing by adding 10 $\mu$L of the tagmentation reaction to a master mix containing 1 $\mu$L of an Illumina i5 and i7 index primer pair mixture, 0.375 $\mu$L of ExTaq polymerase (Takara), 5 $\mu$L of ExTaq buffer, 4 $\mu$L of a dNTP mixture, and 29.625 $\mu$L of sterile molecular biology grade water. To multiplex samples for sequencing, we generated all 96 possible index oligo

combinations using 8 i5 and 12 i7 index primers. Libraries were PCR amplified on a Sim-pliAmp thermal cycler (Thermo Fisher Scientific) using the following parameters: initial dena-turation at 95˚C for 30 seconds, then 17 cycles of 95˚C for 10 seconds (denaturation), 62˚C for 30 seconds (annealing), and 72˚C for 3 minutes (extension). The DNA concentration of each reaction was quantified using the Qubit dsDNA BR assay kit (Thermo Fisher Scientific). We pooled equimolar amounts of each sample, ran this mixture on a 2% agarose gel, and extracted and purified DNA in the 400 bp to 600 bp region using the Monarch Gel Extraction Kit (NEB) according to the manufacturer's instructions.

The pooled library was submitted to the University of Minnesota Genomics Center (UMGC) for quality control assessment and Illumina sequencing. UMGC staff performed three quality control (QC) assays prior to sequencing. The PicoGreen dsDNA quantification reagent (Thermo Fisher Scientific) was used to determine library concentration, with a concentration $\geq 1$ ng/$\mu$L required to pass. The Tapestation electrophoresis system (Agilent Technologies, Santa Clara, CA, USA) was used to determine library size, with libraries in the range of 200 to 700 bp passing. Finally, the KAPA DNA Library Quantification kit (Roche, Basel, Switzerland) was used to deter-mine library functionality, with libraries requiring a concentration $\geq 2$ nM to pass. The submit-ted library passed each QC assay. The library was sequenced on a Next-Seq 550 instrument in mid-output, 75 bp paired-end mode, generating 153,887,828 reads across all samples, with a median of 9,757,090 and a range of 5,994,921 to 14,753,319 reads per sample. The mean read quality for all samples was $> 30$. The median read coverage of the genome was 21, with a range of 16 to 25 across all samples. Raw whole-genome sequencing data was deposited the NIH Sequence Read Archive under Bioproject accession PRJNA885116.

## QTL mapping

We used a previously-described approach to identify QTLs from our whole-genome sequenc-ing data [25, 86, 87]. We initially filtered our raw reads to retain only those with a mean base quality score greater than 30. Filtered reads were aligned to the *S. cerevisiae* reference genome (sacCer3) with the Burroughs-Wheeler alignment tool [161]. We used samtools [162] to first remove unaligned reads, non-uniquely aligned reads, and PCR duplicates, and then to produce vcf files containing coverage and allelic read counts at each of 18,871 high-confidence, reliable SNPs [63, 92], with BY alleles as reference and RM alleles as alternative alleles.

QTLs were called from allele counts using the MULTIPOOL algorithm [163]. MULTI-POOL estimates a logarithm of the odds (LOD) score by calculating a likelihood ratio from two models. In the noncausal model, the locus is not associated with the trait and the high and low proteasome activity pools have the same frequency of the BY and RM alleles. In the causal model, the locus is associated with the trait, such that the BY and RM allele frequencies differ between pools. QTLs were defined as loci with a LOD $\geq 4.5$. In a previous study [25], we empirically determined that this threshold produces a 0.5% false discovery rate (FDR) for TFT-based genetic mapping by bulk segregant analysis. We used the following MULTIPOOL settings: bp per centiMorgan = 2,200, bin size = 100 bp, effective pool size = 1,000. As in previ-ous studies [86, 87], we excluded variants with allele frequencies higher than 0.9 or lower than 0.1 [25, 86, 87]. QTL confidence intervals were defined as a 2-LOD drop from the QTL peak (the position in the QTL interval with the highest LOD value). We computed the RM allele fre-quency difference ($\Delta$AF) between the high and low proteasome activity pools at each allele to visualize QTLs. We also used $\Delta$AF at each QTL peak to determine the magnitude and direction of the QTL's effect. When the RM allele frequency difference at a QTL is positive, the RM allele of the QTL is associated with higher proteasome activity. Negative RM allele frequency differ-ences indicate QTLs where the RM allele is associated with lower proteasome activity. Because

allele frequencies are affected by random counting noise, we smoothed allele frequencies along the genome using loess regression prior to calculating ΔAF for each sample.

## QTL fine-mapping by allelic engineering

We used CRISPR-Cas9 to edit the *RPT6* -175 locus in the BY strain. Guide RNAs (gRNAs) targeting *RPT6* were obtained from the CRISPR track of the UCSC Genome Browser [164]. To control for potential off-target edits by CRISPR-Cas9, we used two unique guide RNAs to engineer each allelic edit. We selected two gRNAs in the *RPT6* open-reading frame (ORF) based on their proximity to the *RPT6* -175 variant (PAM sequences 226 and 194 bp from *RPT6* -175), their CRISPOR specificity scores [165] (100 each, where 100 is the highest possible predicted specificity), and their predicted cleavage scores [166] (66 and 56, where > 55 indicates high predicted cleavage efficiency). We inserted each gRNA into a plasmid that expresses Cas9 under the control of the constitutively active *TDH3* promoter as follows. We digested backbone plasmid BFA0224 [167] with the restriction enzymes HpaI and BsmBI (NEB) to remove the backbone vector's existing gRNA. The cut vector was gel purified using the Monarch Gel Extraction kit (NEB) according to the manufacturer's instructions. We then performed isothermal assembly cloning using the HiFi Assembly Kit (NEB) with the gel purified vector backbone and oligos encoding each gRNA (OFA1198 or OFA1199; S5 Table) to create plasmids BFA0242 and BFA0243 (S3 Table). Plasmids were miniprepped from DH5α *E. coli* cells using the Monarch Plasmid Miniprep kit (NEB). The sequence identities of BFA0242 and BFA0243 were confirmed by Sanger sequencing.

We created repair templates for co-transformation with BFA0242 and BFA0243 as follows. We first extracted genomic DNA from BY and RM using the "10 minute prep" protocol [168]. Genomic DNA from each strain was used as a template for PCR amplification of the *RPT6* promoter using oligos OFA1204 and OFA1207 (S5 Table). To prevent Cas9 cutting after editing of the *RPT6* -175 locus, we introduced two synonymous substitutions into the *RPT6* ORF by converting the serine codons TCC and AGT to TCA at base pairs 22–24 and 49–51, respectively. Synonymous substitutions were introduced using splicing overlap by extension PCR [153] with primers OFA1208 and OFA1209. Full repair templates were then amplified using either the BY or RM *RPT6* promoter and the BY *RPT6* ORF as templates in a splicing overlap extension by PCR reaction with primers OFA1204 and OFA1205 (S5 Table). The sequence identify of all repair templates was verified by Sanger sequencing.

To create BY strains with edited *RPT6* alleles, we co-transformed 150 ng of either plasmid BFA0242 or BFA0243 with 1.5 μg of repair template using the transformation protocol above. The transformation reaction was streaked onto solid SC medium lacking histidine to select for the *HIS3* selectable marker in BFA0242 or BFA0243. Colonies from transformation plates were single-colony purified on solid medium lacking histidine, then patched onto solid YPD medium. To verify allelic edits, we performed colony PCR using oligos 1204 and 1206 (S5 Table). Reaction products were gel purified using the Monarch Gel Extraction kit (NEB) and Sanger sequenced using oligos OFA1204 and OFA1206 to confirm both the sequence of the *RPT6* promoter and the synonymous substitutions in the *RPT6* ORF. Strains with the desired edits were then transformed to contain TFT reporters as indicated above. We tested 12 independent biological replicates per strain per guide RNA per TFT. For subsequent statistical analyses, we pooled strains with the same allelic edit engineered with unique guide RNAs.

We constructed plasmids to overexpress *RPT6* via the gene's native promoter or the strong, constitutively active *ACT1* promoter (S4 Table). To do so, we digested the backbone plasmid BFA001 (Addgene plasmid #35121—a gift from John McCusker) with HindIII and NdeI (NEB). We then PCR amplified the low copy *CEN* origin of replication and the *ACT1*

promoter from plasmid BFA0129. The *RPT6* promoter and open reading frame were amplified from genomic DNA extracted from the BY strain using the "10 minute prep protocol" [168]. We PCR amplified the NatMX [149] cassette, which confers resistance to the antibiotic clonNAT, from BFA001. The mNeon [169] GFP was amplified from plasmid BFA0254. Plasmids were assembled using isothermal assembly cloning (Hifi Assembly Cloning Kit; NEB) and their sequence identity was verified by whole-plasmid sequencing (Plasmidsaurus, Eugene, OR, USA). The overexpression plasmids thus contain the low copy *CEN* replication origin, *RPT6* under the control of the *RPT6* or *ACT1* promoter, and the NatMX resistance cassette. For experiments quantifying Rpt6 abundance, we used plasmids BFA0263 and BFA0264 in which Rpt6 is tagged with mNeon, while for experiments measuring proteasome activity during *RPT6* overexpression, we used plasmids BFA0267 and BFA0268 in which Rpt6 is not tagged with a fluorophore. In parallel, we also constructed an empty vector control plasmid (BFA0271), which contains the HindIII and NdeI digested BFA001 backbone, the *CEN* replication origin, and the NatMX resistance cassette (S3 Table). To tag the chromosomal and plasmid *RPT6* gene copies, we appended mNeon to the protein's N-terminus to avoid interference with the protein's C-terminal tail, which serves an important role in proteasome 19S regulatory particle assembly [142–144]. Previous studies have successfully used N-terminal Rpt6 tagging to measure the protein's abundance and proteasome activity [170–173]. We also tagged the chromosomal *ALG13* open reading frame using a similar procedure. Alg13 contains a C-terminal degron that is necessary for the protein's proteasomal degradation [174]. Because failure to degrade excess *ALG13* produces glycosylation defects [174], we appended the mNeon tag to the protein's N-terminus.

## Data and statistical analysis

All data and statistical analyses were performed using R [156]. In all boxplots, the center line shows the median, the box bounds the first and third quartiles, and the whiskers extend to 1.5 times the interquartile range. DNA binding motifs in the *RPT6* promoter were assessed using the Yeast Transcription Factor Specificity Compendium database [175]. We inferred the allelic status of *RPT6* -175 by comparing the BY and RM alleles to a likely-ancestral Taiwanese strain. The frequency of the RM allele at *RPT6* -175 was calculated across and within clades of a global panel of 1,011 *S. cerevisiae* isolates [115]. Final figures and illustrations were made using Inkscape (version 0.92; Inkscape Project).

## Supporting information

**S1 Table. Proteasome activity QTLs.**
(XLSX)

**S2 Table. Overlap of proteasome activity QTLs with known causal genes for N-end Rule QTLs.**
(XLSX)

**S3 Table. Plasmids.** List of plasmids used in the study.
(XLSX)

**S4 Table. Yeast strains.** List of yeast strains used in the study.
(XLSX)

**S5 Table. Oligonucleotides.** List of oligonucleotides used in the study.
(XLSX)

## Acknowledgments

We thank the members of the Albert laboratory for feedback on the project and manuscript. We thank the University of Minnesota's Flow Cytometry Resource and Genomics Center for their contributions to the project.

## Author Contributions

**Conceptualization:** Mahlon A. Collins, Frank W. Albert.

**Formal analysis:** Mahlon A. Collins.

**Funding acquisition:** Mahlon A. Collins, Frank W. Albert.

**Investigation:** Mahlon A. Collins, Randi Avery.

**Methodology:** Mahlon A. Collins, Frank W. Albert.

**Resources:** Frank W. Albert.

**Supervision:** Mahlon A. Collins, Frank W. Albert.

**Validation:** Mahlon A. Collins, Randi Avery.

**Visualization:** Mahlon A. Collins.

**Writing – original draft:** Mahlon A. Collins.

**Writing – review & editing:** Mahlon A. Collins, Frank W. Albert.

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
