## [Decision Letter · Decision Letter 0]

25 Jul 2022

Dear Dr Collins,

Thank you very much for submitting your Research Article entitled 'Substrate-Specific Effects of Natural Genetic Variation on Proteasome Activity' to PLOS Genetics.

The manuscript was fully evaluated at the editorial level and by independent peer reviewers. The reviewers appreciated the attention to an important problem, but raised some substantial concerns about the current manuscript. Based on the reviews, we will not be able to accept this version of the manuscript, but we would be willing to review a much-revised version. They would like to see additional expriments, particularly around RPT6.

If you decide to revise the manuscript for further consideration at PLOS Genetics, please aim to resubmit within the next 60 days, unless it will take extra time to address the concerns of the reviewers, in which case we would appreciate an expected resubmission date by email to plosgenetics@plos.org.

Please do not hesitate to contact us if you have any concerns or questions.

Yours sincerely,

Michael Snyder, Ph.D.

Academic Editor

PLOS Genetics

Gregory P. Copenhaver

Editor-in-Chief

PLOS Genetics

Reviewer's Responses to Questions

**Comments to the Authors:**

Reviewer #1: Collins et al. have taken advantage of the high genetic diversity between two distinct Saccharomyces cerevisiae yeast strains, BY and RM, and used it with a pair of tandem fluorescent timer (TFT) ubiquitin-independent degron fusions to identify QTLs that affect proteasome-mediated degradation rates of the two TFT reporters. They identify 15 QTLs, 12 of which have substrate-specific effects. They use this information together with data from a highly similar study (also deposited in bioRxiv by the same group) to identify potential genes responsible for the QTL effects. One possible hit is in the intergenic region upstream of RPT6, which encodes a regulatory subunit of the proteasome. They identify a single nucleotide polymorphism and using CRISPR/Cas9 to change this residue in a BY background to the RM base pair and find that this is sufficient to moderately increase degradation of a number of the reporters (N-end rule and one of the ubiquitin-independent reporters).

This is an interesting study as far as it goes, and the experimental analysis is solid overall. I confess to being a little disappointed that it did not fully follow through on at least the apparent upstream RPT6 hit. One could also argue that this manuscript and the aforementioned bioRxiv paper should be combined, especially since data from the latter paper are used extensively here, but this is not my call.

I found the high fraction of substrate-specific QTLs a little surprising. Among the 15 QTLs, I would expect multiple proteasome subunit or assembly factor hits, and it is hard to see how the proffered explanation for the 12 substrate-specific ones –that recognition or binding of these two substrates involves different proteasome subunits– can fully account for such specificity. The fact that MKT1 and IRA2 were likely hits makes one think that many/most of the QTLs will turn out to affect proteasomal degradation in highly indirect ways, which while in all likelihood are real effects, will be difficult to understand more deeply.

Experimentally, it should be straightforward to examine the RPT6 hit more carefully using the CRISPR’ed allele created. First, does the SNP really affect only RPT6 transcription and not the nearby ALG13 gene, to which it is actually closer? The latter gene is involved in an essential biological process, protein glycosylation, and indirect effects from changes of its expression could certainly impact the protein homeostasis network (for example, endoplasmic reticulum-associated protein degradation). Second, the authors make the interesting speculation that because the RM strain is from a “European wine” lineage, enhanced proteasome activity might be selected for its possibly enhanced ethanol tolerance. This could be tested experimentally with the CRISPR’d and matched BY strain too. Third, the SNP is proposed to allow Yap1 binding in the RM but not BY strain. The dependence of increased RPT6 mRNA on Yap1 in the CRISPR’d strain can be measured. Finally, if the SNP increases Rpt6 levels, the prediction of the authors is that there will be increased proteasomes as a result. This is also testable (although it might be beyond the expertise of a genetics-intensive lab, so I would not demand this). In summary, it would be important, in my opinion, to show that at least one QTL (RPT6-175) directly affects proteasome activity or regulation in a way that can be understood at least roughly in mechanistic terms.

Some additional comments:

Line 41 “Ubiquitin system enzymes bind degradation-promoting signal sequences” – generally, only the E3 enzymes do.

Line 57: “Until recently, it was largely unknown how individual genetic differences affect UPS protein degradation” There are lots of genetic data on genetic differences that affect the UPS. I think what is meant here is the question of how natural or semi-natural genetic variation affects the UPS. This should be clarified.

Line 88: “proteasome can exist in multiple configurations” What exactly is meant by this phrase? Please define ‘configuration’.

Line 99: “when using these [ubiquitin reporter] systems to map genetic influences on UPS activity, variant effects on the ubiquitin system may mask or obscure specific effects on proteasomal protein degradation.” Not clear why this should be the case. For example, if a UPS substrate is used as a reporter, why would variants in proteasome subunits not be detected? I would expect to see both ubiquitin conjugation enzyme and proteasome allelic variations unless the latter specifically have a lower relative impact. Please clarify.

Line 216: “Ac/N-degrons are generated and recognized by a common set of molecular effectors. Reflecting this, many QTLs for Ac/N-degrons affect all or a majority of the full set of Ac/N-degrons. By contrast, Arg/N-degrons are created and recognized via molecular mechanisms that affect individual or small subsets of Arg/N-degrons.” I don't follow this reasoning: each class requires a common set of factors (E1, E2s, E3s) and has a small number of processing enzymes specific to certain degrons: for Ac/N-degrons, some will require MAPs (not sure how the N-end substrates were made for this study though) and different ones will require different Ac transferases, while for Ac/N-degrons, a very small number will require a deamidase and a small number will require Arg transferase.

Line 264: “multiple factors specifically regulate the degradation of ubiquitin-independent proteasomal substrates, without affecting the degradation of ubiquitinated substrates (ref. 80).” Please give an example or two.

Line 318: “The Rpn4 degron, in particular, is recognized by distinct 19S regulatory particle receptors from the other substrates tested here (Ref. 64) and may, therefore, be unaffected by RPT6 -175.” Ref. 64 only tested the Rpn4 degron; similar mapping has not been done for ODC degron binding to my knowledge. It could be that the ODC degron is bound to the same subunits.

Line 421: “Ref 97 showed overexpression of alpha1, not alpha3/Pre9, has these effects. There is also work in human cells showing alpha4/PSMA7 overexpression can enhance proteasome activity and lead to enhanced heavy metal resistance (Padmanabhan et al. 2016).

Line 427: “Thus, PAAF1 association with Rpt6 creates a stable Rpt6 pool that can be used to rapidly drive proteasome assembly.” This doesn't really address the question of how increasing levels of one subunit of the ~30+ subunit proteasome complex drives assembly of the whole particle. There are some examples of how this can work, but none involve Rpt6 to my knowledge.

Reviewer #2: In this manuscript Collins and coworkers examine the influence of natural genetic variation in yeast on the stability of two degrons (as a proxy for proteasome activity). To do so they leverage fluorescence protein timers, where steady-state ratiometric measurements of fluorescence are related to the stability of each reporter. Using flow cytometry they perform bulk segregant analysis and map over a dozen loci that are linked to differences in reporter stability. Many such linkages appear to selectively influence one reporter but not the other. Finally, using CRISPR-Cas9 approaches, the authors demonstrate that one of the alleles they identified can recapitulate the effects they identified statistically. Overall this is a fine manuscript and the data seem sound. Additional experiments, analyses, and qualifications are needed (described below) to support or clarify specific claims. Likewise, there are multiple opportunities to improve impact and generalizability.

**Major comments**

1. **Controls for the assay.** A couple of points are important to discuss or test explicity. First, as it is present in *the* key transcriptional regulator of the proteosome, does the present of additional Rpn4 degron copies influence the UPS? Likewise, controls for growth phase, cell shape, size, adhesion, and other properties that might influence turnover are important. It may not be possible to fully account for all confounding variables, but they should at least be commented upon.

2. **Comparison to prior literature for these degrons.** The FP timers enable single cell resolution and impressive scale. At the same time, they report on steady-state fluorescent ratios rather than actual kinetic proteasome function. How do the differences between the OTC and RPN4 degrons measured here compare to conventiaonl measurements of their proteasomal turnover? Likewise, do the ratios obtained with this assay behave as one might expect when titrating proteasome inhibitors etc.? Finally, it is critical that the manuscript include some discussion of the quantitative ranges of difference in degradation rates that can be observed, and an estimation of those that cannot, based on the maturation kinetics of the GFP and RFP variants used.

3. **Comparison to known genetic architecture of the UPS and other core regulatory processes.** Others have examined UPS function with deletion and overexpression experiments, among other approaches. How do the results here compare in terms of number of genes identified, overlap, etc.? Do the authors find essential genes that were not previously seen? Did prior studies also find substrate specific genetic contributions? And how do the number and effect sizes of QTLs here compare to those seen for other traits examined by MPRA by both the PI and others in the field?

4. **Completeness of the genetic dissection and origins of degron specific differences.** How much of the difference between BY and RM can be explained by the loci that the authors identified? Also, how much of the specificity in mapping for each degron is a simple property of their different stabilities to begin with? From the investigator's prior work with N-end rule reporters and GFP fusions it might be possible (and interesting) to speculate on how, quantitatively, this type of regulation may integrate with other types of trans regulatory variation.

5. **Tie up the mechanism.** I commend the allele reconstruction experiment, but it also left me wondering about the proposed mechanism, which is presently based almost entirely on speculation. First, although altered regulation of Rpt6 is obviously the easier explanation, the authors should do an experiment to exclude Alg13. Second, seeing allele-specific rescue with Yap1 overexpression or increased Rpt6 ovexpression would signficantly strengthen their case. Of course there can be challenges with such experiments, but they are simple and well worth trying.

6. **Ecological impact.** The authors determine that the RM allele is derived, which seems reasonable. But at the same time in the tree in Fig. S1, which I would recommend moving to the main text, there are several niches that also have a high frequency of the allele. It may challenging to distinguish positive from balancing selection, but it does look like there may be some enrichment for fermentative niches? I would recommend that the authors investigate this possibility.

**Minor comments**

7. In my view, Figure 1 can probably go to the supplement, whereas Figure S1 should probably be in the main text.

8. For all quantitative statements I would ask the authors to provide not only the p-value and statisical test used but also the magnitude of the effect.

9. What happened with the outlier in Fig. 2E?

10. Limitations and advatnages of the BSA approach should be discussed more fully. For example epistasis might be very hard to see, etc.

11. Can the authors please add columns for the number of genes and polymorphisms within each confidence interval described in Table 1?

**Have all data underlying the figures and results presented in the manuscript been provided?**

Reviewer #1: Yes

Reviewer #2: Yes

PLOS authors have the option to publish the peer review history of their article (what does this mean?). If published, this will include your full peer review and any attached files.

Reviewer #1: No

Reviewer #2: No

---

## [Editor Report · Decision Letter 1]

4 Apr 2023

Dear Dr Collins,

We are pleased to inform you that your manuscript entitled "Substrate-Specific Effects of Natural Genetic Variation on Proteasome Activity" has been editorially accepted for publication in PLOS Genetics. Congratulations!

Before your submission can be formally accepted and sent to production you will need to complete our formatting changes, which you will receive in a follow up email. Please be aware that it may take several days for you to receive this email; during this time no action is required by you. Please note: the accept date on your published article will reflect the date of this provisional acceptance, but your manuscript will not be scheduled for publication until the required changes have been made. Please also make sure all data are submitted to public repositories.

Yours sincerely,

Michael Snyder, Ph.D.

Academic Editor

PLOS Genetics

Gregory Copenhaver

Editor-in-Chief

PLOS Genetics

Comments from the reviewers (if applicable):

**Data Deposition**

http://datadryad.org/submit?journalID=pgenetics&manu=PGENETICS-D-22-00705R1

**Press Queries**

---

## [Editor Report · Acceptance letter]

26 Apr 2023

PGENETICS-D-22-00705R1 

Substrate-Specific Effects of Natural Genetic Variation on Proteasome Activity 

Dear Dr Collins, 

We are pleased to inform you that your manuscript entitled "Substrate-Specific Effects of Natural Genetic Variation on Proteasome Activity" has been formally accepted for publication in PLOS Genetics! Your manuscript is now with our production department and you will be notified of the publication date in due course.

With kind regards,

Anita Estes

PLOS Genetics

On behalf of:
